# Understanding the evolution of multiple drug resistance in structured populations

**David V McLeod\*, Sylvain Gandon\***

Centre D'Ecologie Fonctionnelle & Evolutive, CNRS, Univ Montpellier, EPHE, IRD, Montpellier, France

**Abstract** The evolution of multidrug resistance (MDR) is a pressing public health concern. Yet many aspects, such as the role played by population structure, remain poorly understood. Here, we argue that studying MDR evolution by focusing upon the dynamical equations for linkage disequilibrium (LD) can greatly simplify the calculations, generate more insight, and provide a unified framework for understanding the role of population structure. We demonstrate how a general epidemiological model of MDR evolution can be recast in terms of the LD equations. These equations reveal how the different forces generating and propagating LD operate in a dynamical setting at both the population and metapopulation levels. We then apply these insights to show how the LD perspective: (i) explains equilibrium patterns of MDR, (ii) provides a simple interpretative framework for transient evolutionary dynamics, and (iii) can be used to assess the consequences of different drug prescription strategies for MDR evolution.

**\*For correspondence:**
david.mcleod@cefe.cnrs.fr (DVM);
sylvain.gandon@cefe.cnrs.fr (SG)

**Competing interests:** The authors declare that no competing interests exist.

## Introduction

Antibiotic resistance is one of the biggest current public health problems, with antibiotic resistant infections responsible for tens of thousands of deaths annually (*O'Neill, 2015*). Of particular concern is the evolution of *multidrug resistant* (MDR) pathogens, that is, pathogens resistant to multiple classes of antibiotics. Despite its importance, understanding the evolution of MDR remains an ongoing challenge, as it is typically not captured by our understanding of the evolution of single drug resistance (for which there is a large body of theory; e.g., *Blanquart, 2019*; *Bonhoeffer et al., 1997*; *Lipsitch et al., 2000*; *Bergstrom et al., 2004*; *Austin and Anderson, 1999*). For instance, suppose we have two drugs, $A$ and $B$, and that a fraction $f_{AB}$ of infections caused by the pathogen of interest are resistant to both drugs. To understand MDR evolution, we need to understand what determines the frequency $f_{AB}$. If $f_A$ and $f_B$ are the frequency of infections resistant to drugs $A$ and $B$, and $D$ denotes any non-random association between resistance to drugs $A$ and $B$, then

$$f_{AB} = f_A f_B + D. \qquad (1)$$

If $D = 0$, then the evolution of resistance to each drug is independent, and so multiple drugs do not qualitatively alter the evolutionary dynamics of single drug resistance. However, whenever $D \neq 0$, understanding the fitness costs and benefits of resistance to each drug in isolation is insufficient to understand the evolution of MDR, because doing so will not tell us what factors govern the propagation of $D$, which in turn will affect $f_A$ and $f_B$. Thus the challenge of understanding MDR evolution can be recast as understanding the dynamics of $D$. The quantity $D$ is referred to as *linkage disequilibrium* (LD), and it has been extensively studied in population genetics (e.g. *Lewontin, 1964*; *Felsenstein, 1965*; *Ohta, 1982a*; *Barton, 1995*; *Rice, 2004*; *Slatkin, 2008*), particularly as it relates to population structure (*Ohta, 1982b*; *Slatkin, 1975*; *Li and Nei, 1974*; *Nei and Li, 1973*; *Lenormand and Otto, 2000*; *Martin et al., 2006*). However, there has been little attempt to apply these insights to MDR evolution; often the dynamics of doubly resistant infections are neglected to

 

simplify the analysis of single drug resistance (e.g. *Bergstrom et al., 2004*; *Bonhoeffer et al., 1997*; *Beardmore et al., 2017*).

Here, we consider a simple epidemiological model of a primarily asymptotically carried pathogen (e.g. *Staphylococcus* spp. or *Enterococcus* spp.) in a structured host population. We show how this model relates to general dynamical equations for LD (*Day and Gandon, 2012*), in turn revealing the role of population structure in MDR evolution. We then use these equations to show how analyzing problems from the LD perspective: (i) reveals the evolutionary logic underlying patterns of MDR at equilibrium, which we use to build on a recent paper on MDR evolution (*Lehtinen et al., 2019*); (ii) provides a framework for understanding transient evolutionary dynamics; and (iii) provides insight on the consequences different drug prescription strategies have on MDR, which we apply to a hospital-community setting.

## Results

In what follows we will introduce and analyze a model of MDR evolution. We will highlight the most important aspects here while providing more extensive details in the Materials and methods 'Model derivation'. All notation used is summarized in *Table 1*.

Consider an asymptotically carried pathogen in a metapopulation consisting of $N$ host populations in which two drugs $d$ are prescribed, specifically, drug $d = A$ and drug $d = B$. Focus upon population $x$. Let $S^x$ and $I_{ij}^x$ denote the density of susceptible hosts and $ij$-infections, respectively, at time $t$, where $i$ indicates if the infection is resistant ($i = A$) or not ($i = a$) to drug $A$ and $j$ indicates if the infection is resistant ($j = B$) or not ($j = b$) to drug $B$. Susceptible hosts contract $ij$-infections at a per-capita rate $\beta_{ij}^x I_{ij}^x$, where $\beta_{ij}^x$ is a rate constant, while $ij$-infections are naturally cleared at a per-capita rate $\alpha_{ij}^x$. Hosts are treated with drugs $A$, $B$, or both in combination at per-capita rates $\tau_A^x$, $\tau_B^x$, and $\tau_{AB}^x$, respectively. Treatment is instantaneous and resistance is complete, that is, if the host that receives treatment is infected by a strain sensitive to the drug, the infection is cleared instantaneously, whereas if the host that receives treatment is infected by a strain resistant to the drug, treatment has no effect. Hosts move from population $x$ to $y$ at a per-capita rate $m^{x \to y}$. Transmission between infected hosts leads to superinfection with probability $\sigma$ in which either strain is equally likely to instantaneously outcompete the other (*Nowak and May, 1994*; *Alizon, 2013*). We therefore do not

**Table 1.** Notation used in main text.

In all cases, a quantity indexed with a superscript $x$ is the population $x$ quantity, whereas the absence of a superscript $x$ implies the quantity is for the metapopulation.

| Symbol | Description |
|---|---|
| $I_{ij}^x$ | Density of $ij$-infections in population $x$, where $i = A$ (resp. $i = a$) if infection is resistant (resp. sensitive) to drug $A$ and $j = B$ (resp. $j = b$) if infection is resistant (resp. sensitive) to drug $B$. |
| $I^x$ | Density of total infections in population $x$. |
| $f_d^x, \bar{f}_d$ | Frequency of infections resistant to drug $d$ in population $x$ and the metapopulation, respectively. |
| $D^x, \bar{D}, D_M$ | Linkage disequilibrium (LD) in population $x$, average LD across populations and metapopulation LD, respectively. |
| $m^{x \to y}$ | Per-capita rate at which hosts migrate from population $x$ to $y$. |
| $r^x, \bar{r}$ | Per-capita growth rate of sensitive infections in population $x$ (or 'baseline' per-capita growth rate) and average across populations, respectively. |
| $s_d^x, \bar{s}_d$ | Additive selection coefficient for resistance to drug $d$ in population $x$ and average selection across populations, respectively. |
| $s_E^x, \bar{s}_E$ | Epistasis in fitness across drug resistance loci in population $x$ and average across populations, respectively. |
| $\phi \mu_{ij}^x, \phi \rho_{ij}^x$ | Net change in $ij$-infections in population $x$ due to mutation or recombination, respectively. |
| $\mu_i^x, \bar{\mu}_i$ | Per-capita rate at which mutations generate allele $i$ in population $x$ and average across populations, respectively. |
| $\rho_i^x, \bar{\rho}_i$ | Per-capita rate at which recombination leads to gain of allele $i$ in population $x$ and average across populations, respectively. |
| $s^x, \bar{s}$ | Average selection for drug resistance in population $x$ and average across populations, respectively. |
| $\text{cov}(X, Y)$ | Covariance between the variables $X$ and $Y$, that is, $\text{cov}(X, Y) = \mathbb{E}[XY] - \mathbb{E}[X]\mathbb{E}[Y]$, where $\mathbb{E}[X]$ denotes the expectation of quantity $X$. |
| $\text{coskew}(X, Y, Z)$ | Coskewness between the quantities $X, Y, Z$, that is, $\text{coskew}(X, Y, Z) = \mathbb{E}[(X - \mathbb{E}[X])(Y - \mathbb{E}[Y])(Z - \mathbb{E}[Z])]$. |

allow for prolonged coinfection (Materials and methods 'Model derivation'). Finally, individual infections acquire allele $i$ through either mutation or recombination (during superinfection) at per-capita rates $\mu_i^x$ and $\rho_i^x$, respectively (note that $\rho_i^x$ depends upon infection densities, see Materials and methods *Equation (13)*).

From these epidemiological assumptions, the change in $ij$-infections in population $x$ can be written as the sum of four processes

$$\frac{\mathrm{d}I_{ij}^x}{\mathrm{d}t} = \overbrace{(r^x + \mathbf{1}_A s_A^x + \mathbf{1}_B s_B^x + \mathbf{1}_A \mathbf{1}_B s_E^x)I_{ij}^x}^{\text{per-capita growth}} + \overbrace{\phi\mu_{ij}^x}^{\text{mutation}} + \overbrace{\phi\rho_{ij}^x}^{\text{recombination}} + \overbrace{\sum_{y=1}^{N}(m^{y\to x}I_{ij}^y - m^{x\to y}I_{ij}^x)}^{\text{migration}}, \quad (2)$$

where $\mathbf{1}_d$ is equal to 1 if the $ij$-infection is resistant to drug $d$ and 0 otherwise (e.g., if $ij = AB$, then the per-capita growth is $r^x + s_A^x + s_B^x + s_E^x$) and $\phi\mu_{ij}^x$ and $\phi\rho_{ij}^x$ denote the net change in $ij$-infections due to mutation and recombination (*Figure 1*; Materials and methods *Equations (11) and (13)*). To faciliate comparison with previous results, we have broken the per-capita growth term into four components: the 'baseline' per-capita growth rate, $r^x$, the (additive) selection coefficients for resistance to drugs $A$ and $B$, $s_A^x$ and $s_B^x$, and any epistatic interactions, $s_E^x$. These latter terms have the standard interpretation. If $s_A^x > 0$ (resp. $s_B^x > 0$), then resistance to drug $A$ (resp. $B$) is selected for. If $s_E^x > 0$, there is positive epistasis, and the per-capita growth rate of doubly-resistant infections is greater than would be expected by consideration of the per-capita growth rate of singly-resistant infections. Thus although *Equation (2)* is derived from a specific model, the partitioning is very general and applies to many epidemiological scenarios. We stress that any of the terms $s_d^x$, $s_E^x$, $\phi\mu_{ij}^x$, and $\phi\rho_{ij}^x$ may themselves depend upon population densities (see *Figure 1* for a concrete example). Note that this

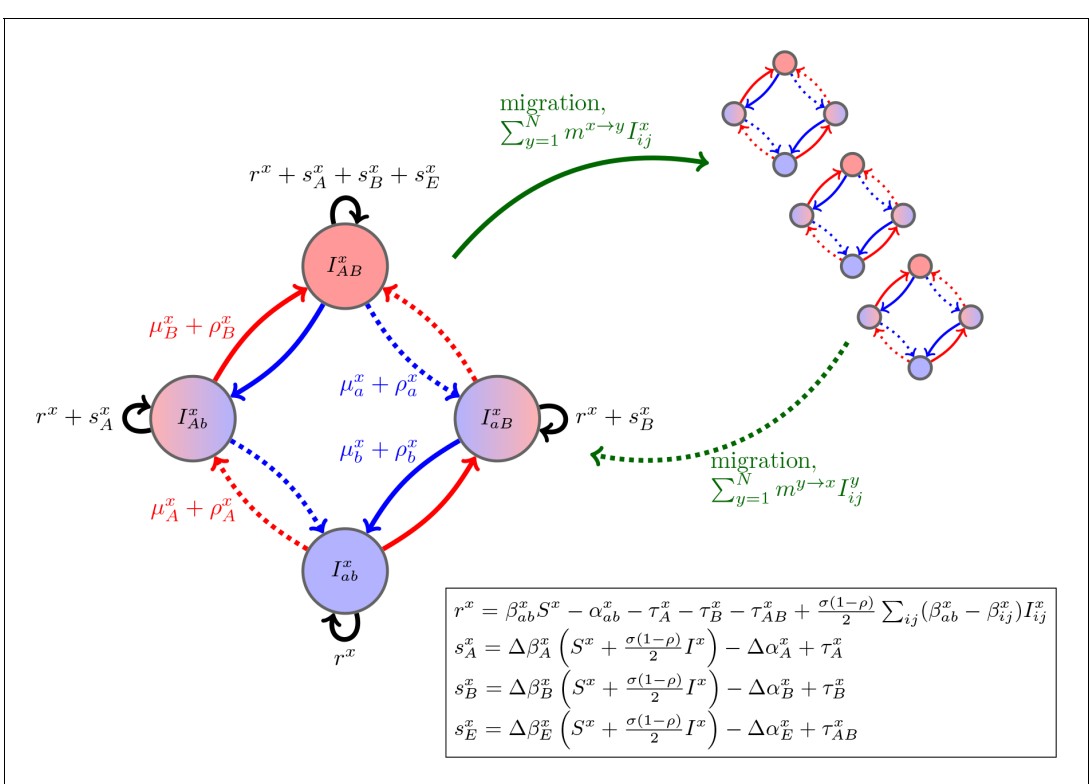

**Figure 1.** Schematic of the dynamics of system (2). The metapopulation consists of $N$ connected populations. Each population has four possible types of infections, linked by one-step mutation or recombination (blue and red arrows), whose per-capita rates are independent of genetic background. The 'baseline' per-capita growth rate of sensitive infections is $r^x$, the additive selection coefficients for drug $A$ and $B$ resistance are $s_A^x$ and $s_B^x$, respectively, while $s_E^x$ denotes any epistatic interactions. In the inset, we compute these quantities for the specific model introduced in the main text, using the notation that $\Delta z_d^x$ and $\Delta z_E^x$ are the contribution of trait $z$ to the additive selection coefficient (for resistance to drug $d$) and to epistasis, respectively, in population $x$ (e.g., $\Delta\beta_A^x = \beta_{Ab}^x - \beta_{ab}^x$ and $\Delta\beta_E^x = \beta_{AB}^x - \beta_{ab}^x - \Delta\beta_A^x - \Delta\beta_B^x$).

partitioning is not arbitrary, particularly as it applies to the selection coefficients and epistasis. The additive selection coefficients and epistasis are defined in terms of their effect upon fitness. In continuous time models, fitness is per-capita growth. Thus, the selection coefficient for allele $k$ measures the additive contribution of allele $k$ to fitness, while epistasis measures the excess of the fitness of strain $AB$ over its value if fitness were additive across the two loci (e.g. *Felsenstein, 1965*; *Karlin, 1975*; *Rice, 2004*; *Kouyos et al., 2009*) (see also *Box 1*). Because epistasis is defined in terms of fitness, how costs of resistance are modeled will typically have implications for whether epistasis occurs or not; for example, multiplicative costs will generate epistasis (*Box 1*; Materials and methods 'Equilibrium analysis of metapopulation consisting of independent populations'). We will return to this point in the examples.

While system (2) contains all the information necessary to analyze MDR evolution, as currently written it is particularly opaque for providing insight. Therefore, we would like to transform it to a form which brings to the forefront the different factors that promote or impede MDR evolution; the way to do this is by focusing upon the dynamical equations for linkage disequilibrium (LD) (*Day and Gandon, 2012*; *Slatkin, 2008*). However, the inclusion of multiple populations means that doing so is not as simple as *Equation (1)* would suggest since there are different scales at which LD and MDR can be measured. As the scale which is of most interest will depend upon the specifics of the

## Box 1. Costs of resistance, epistasis, and multidrug resistance.

The spread of multidrug resistance (MDR) is driven by selection acting on each drug resistance locus, but also on the linkage disequilibrium (LD), which can be produced by epistasis in fitness. Epistasis measures the interaction between resistance alleles (mutations) at different loci and is defined in terms of the per-capita growth rates of different genotypes as:

$$s_E^x \equiv r_{AB}^x + r_{ab}^x - r_{Ab}^x - r_{aB}^x.$$

Selection at each locus, e.g. $s_A^x = r_{Ab}^x - r_{ab}^x$, depends on the effects of the mutations on the phenotypic traits of the pathogen. However, non-additive interactions among these mutations can create epistasis (see inset in *Figure 1*). To better see how these non-additive effects can emerge, consider the costs of drug resistance on pathogen transmission. Let $c_{\beta_d}^x$ denote the parameter controlling the cost of resistance to drug $d$ in population $x$. Then using the notation of *Figure 1*.

| | Transmission rates | | | Epistasis |
|---|---|---|---|---|
| | $\beta_{Ab}^x$ | $\beta_{aB}^x$ | $\beta_{AB}^x$ | $\Delta\beta_E^x$ |
| Additive | $\beta_{ab}^x - c_{\beta_A}^x$ | $\beta_{ab}^x - c_{\beta_B}^x$ | $\beta_{ab}^x - c_{\beta_A}^x - c_{\beta_B}^x$ | $0$ |
| Multiplicative | $\beta_{ab}^x(1 - c_{\beta_A}^x)$ | $\beta_{ab}^x(1 - c_{\beta_B}^x)$ | $\beta_{ab}^x(1 - c_{\beta_A}^x)(1 - c_{\beta_B}^x)$ | $\beta_{ab}^x c_{\beta_A}^x c_{\beta_B}^x$ |

Hence, only multiplicative costs generate non-additive interactions between loci on transmission, $\Delta\beta_E^x$, which leads to epistasis (inset of *Figure 1*); and, in turn epistasis produces LD which affects MDR evolution.

Of course, the magnitude of drug resistance costs and the interaction between these costs at multiple loci need not be additive nor multiplicative. Subject to appropriate constraints on the choice of costs (e.g. $0 \leq \beta_{ij}^x \leq \beta_{ab}^x$), our general framework can account for any pattern of epistasis (see *Figure 1*). The important point is that since epistasis is a key contributor to multilocus evolution, understanding when it occurs and what is producing it (in this case, assumptions about the cost of resistance) can provide valuable insight into the evolutionary dynamics of MDR.

problem, in what follows we will consider MDR evolution at both the population- and metapopulation-level.

## Population-level multidrug resistance

To understand MDR evolution in a given population, say $x$, we need to understand the dynamics of the frequency of infections resistant to drug $A$ and $B$, $f_A^x$ and $f_B^x$, and the dynamics of population LD, $D^x$. First, consider the dynamics of $f_A^x$ (*mutatis mutandis* $f_B^x$). Using *Equation (2)*, it is straightforward to compute

$$\frac{df_A^x}{dt} = \underbrace{s_A^x f_A^x (1-f_A^x)}_{\text{direct selection}} + \underbrace{s_B^x D^x}_{\text{indirect selection}} + \underbrace{s_E^x f_A^x (1-f_A^x)\frac{f_{AB}^x}{f_A^x}}_{\text{epistasis}} +$$
$$\underbrace{(\mu_A^x + \rho_A^x)(1-f_A^x) - (\mu_a^x + \rho_a^x)f_A^x}_{\text{mutation and recombination}} - \underbrace{\sum_{y=1}^{N} m^{y\to x}\frac{I^y}{I^x}(f_A^x - f_A^y)}_{\text{migration}}. \tag{3}$$

where $I^x$ is the total density of infections in population $x$ and $f_{AB}^x = D^x + f_A^x f_B^x$ is the frequency of doubly-resistant infections. A related formulation to *Equation (3)* can be found in *Day and Gandon, 2012* (see also *Rice, 2004*).

*Equation (3)* is partitioned into recognizable quantities. First, if resistance to drug $A$ is selectively advantageous, $s_A^x > 0$, then drug $A$ resistance will increase due to direct selection whose strength is dictated by the genetic variance at the locus, $f_A^x(1-f_A^x)$ (*Fisher, 1930*). Second, if doubly-resistant infections are over-represented in the population, $D^x > 0$, and resistance to drug $B$ is selected for, $s_B^x > 0$, then drug $A$ resistance will increase due to indirect selection upon resistance to drug $B$. Third, if epistasis is positive, $s_E^x > 0$, and there is genetic variance at the locus, drug $A$ resistance will increase due to the disproportionate growth of doubly-resistant infections. Fourth, mutation and recombination will increase drug $A$ resistance when there is a mutation or recombination bias toward gain of drug $A$ resistance, $\mu_A^x > \mu_a^x$ or $\rho_A^x > \rho_a^x$, and the frequency of infections sensitive to drug $A$ exceeds the frequency of infections resistant to drug $A$, $1 - f_A^x > f_A^x$. Finally, migration acts to reduce differences between populations.

It follows that drug $B$ treatment alters the predicted dynamics of resistance to drug $A$ via two main effects: (i) the influence of epistasis and (ii) indirect selection on resistance to drug $B$ mediated through the presence of LD ($D^x \neq 0$). Thus, consider the dynamics of $D^x$,

$$\frac{dD^x}{dt} = \underbrace{(s_A^x - s^x + s_B^x - s^x)D^x}_{\text{selection}} - \underbrace{(\mu^x + \rho^x)D^x}_{\text{mutation and recombination}} + \underbrace{s_E^x f_{AB}^x f_{ab}^x}_{\text{epistasis}} - \underbrace{\sum_{y=1}^{N} m^{y\to x}\frac{I^y}{I^x}\left(D^x - D^y - (f_A^x - f_A^y)(f_B^x - f_B^y)\right)}_{\text{migration}}, \tag{4}$$

where $s^x = f_A^x s_A^x + f_B^x s_B^x + f_{AB}^x s_E^x$ is the average selection for resistance, $f_{ab}^x = 1 - f_A^x - f_B^x + f_{AB}^x$ is the frequency of doubly-sensitive infections, and $\mu^x$ and $\rho^x$ are the total per-capita rates of mutation and recombination, respectively (e.g. $\mu^x = \mu_a^x + \mu_A^x + \mu_b^x + \mu_B^x$; Materials and methods 'Model derivation').

*Equation (4)* is partitioned into four key processes. First, excess selection for resistance to drug $A$ (resp. $B$), $s_A^x - s^x$, can cause pre-existing LD ($D^x \neq 0$) to increase or decrease. For example, if $s_A^x > s^x$ and $D^x > 0$ then LD will increase. This is because drug $A$ resistant infections are fitter than the average resistant infection and so will increase in frequency. If $D^x > 0$, it is more likely this increase will occur in doubly-resistant infections, thereby increasing $D^x$. Second, mutation and recombination removes any LD present at a rate proportional to the LD (*Rice, 2004*; *Slatkin, 2008*). Third, epistasis generates same-sign LD, that is, positive epistasis, $s_E^x > 0$, leads to MDR over-representation, $D^x > 0$ (*Felsenstein, 1965*; *Lewontin and Kojima, 1960*; *Lewontin, 1964*). Positive epistasis could occur if double-resistance costs are less than expected (*Trindade et al., 2009*; *MacLean et al., 2010*; *Hall and MacLean, 2011*) or drugs are prescribed in combination (*Bretscher et al., 2004*; *Day and Gandon, 2012*).

Migration is the final term of *Equation (4)* and reveals how the metapopulation structure affects population LD. Like epistasis, migration does not require preexisting LD to operate on LD (*Li and Nei, 1974*; *Slatkin, 1975*; *Feldman and Christiansen, 1974*; *Ohta, 1982a*; *Ohta, 1982b*). In

particular, LD in population $x$ will be generated whenever the frequencies of resistance to drugs $A$ and $B$ differ between population $x$ and any other connected population, say $y$. If both types of resistance are more common in one population than the other $(f_A^x - f_A^y)(f_B^x - f_B^y) > 0$, then migration will generate positive LD in both populations, $D^x > 0$ and $D^y > 0$. If instead drug $A$ resistance is more prevalent in one population, while drug $B$ resistance is more prevalent in the other, migration will generate negative LD in both populations.

Notice the presence of the multiplier $I^y/I^x$ in the final term of **Equation (4)**. If the populations have roughly the same density of infections, then this term is unimportant. However, when one population, say $y$, has much fewer total infections than population $x$, $I^y \ll I^x$, the term $I^x/I^y$ will be very large, whereas $I^y/I^x$ will be very small. Consequently, the ability of migration to propagate LD will be greater in population $y$ than $x$, and so all else being equal we would predict the population with a lower density of infections will have a greater magnitude of LD than the population with a higher density of infections.

The next insight shows the importance of also taking into account **Equation (3)**. In particular, if we only inspected the migration term of **Equation (4)** we might conclude that as the per-capita migration rate, $m^{y \to x}$, increases, so too will the ability of migration to propagate LD. However, the magnitude of population LD is actually maximized at intermediate migration rates (**Figure 2**). The reason is because the quantity $m^{y \to x}$ has two effects. On the one hand, it directly multiplies the

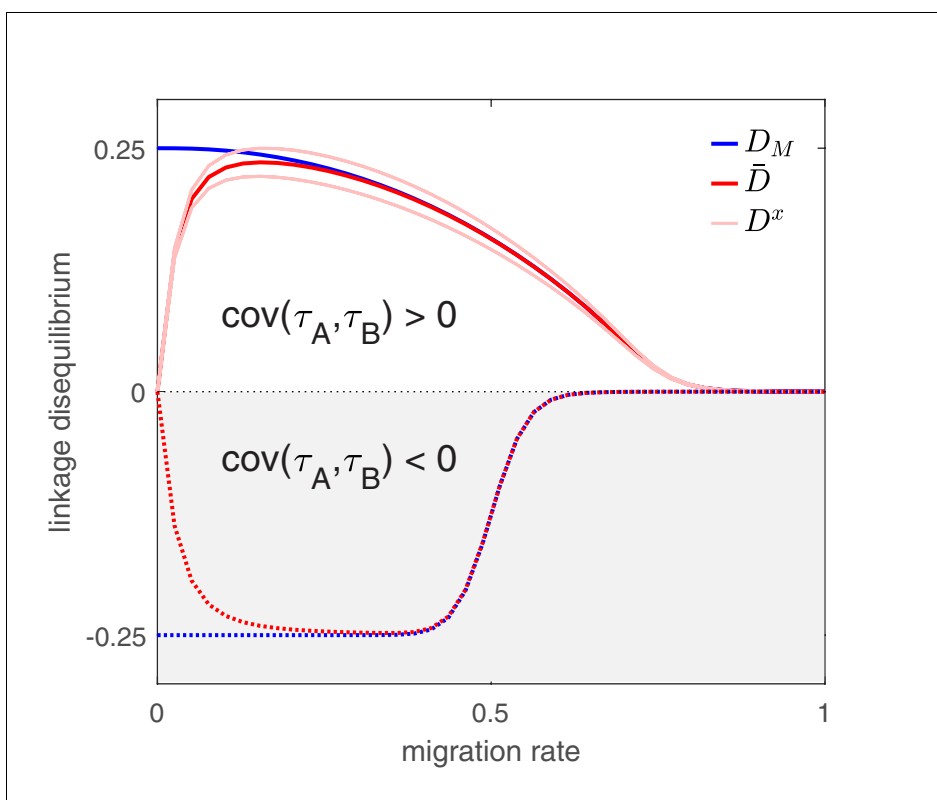

**Figure 2.** The effect of migration upon LD at equilibrium depends upon the scale at which LD is measured. Here, we show equilibrium LD in a metapopulation consisting of four populations. Two scenarios are shown. In the first scenario (solid lines), drug $A$ and drug $B$ are both prescribed in the same two populations while the other two populations receive no drugs, thus $\mathrm{cov}(\tau_A, \tau_B) > 0$; this yields $\mathrm{cov}(f_A, f_B) > 0$ and so positive population, average, and metapopulation LD, that is, $D^x, \bar{D}, D_M > 0$. In the second scenario (dashed lines), drug $A$ is prescribed in two populations and drug $B$ is prescribed in the other two populations, thus $\mathrm{cov}(\tau_A, \tau_B) < 0$; this yields $\mathrm{cov}(f_A, f_B) < 0$ and negative population, average, and metapopulation LD, i.e., $D^x, \bar{D}, D_M < 0$. Because we assume identical treatment rates and costs of resistance for either drug, in the second scenario all the populations have the same LD, whereas in the first scenario, since the drugs are prescribed unequally across populations, the LD observed in each of the two pairs of populations diverge. Specifically, populations experiencing greater selection due to increased drug prescription also have greater LD; this follows from the first term in **Equation (4)**.

migration term in *Equation (4)* thereby magnifying migration's potential role in LD build-up, while on the other hand, it also balances infection frequencies between populations (*Equation (3)*), which in turn will reduce the magnitude of $(f_A^x - f_A^y)(f_B^x - f_B^y)$ in *Equation (4)*. These conflicting forces mean the magnitude of population LD tends to be maximized when migration is neither too infrequent nor too frequent (*Figure 2*).

## Metapopulation-level multidrug resistance

Now what happens to LD and MDR evolution at the metapopulation-level? Here we will use $\bar{X}$ to denote the metapopulation average of quantity $X^x$, e.g., $\bar{f}_A$ is the average drug $A$ resistance in the metapopulation (see Materials and methods 'Metapopulation LD and MDR' for further details). Using this notation, then analogously to the population case, metapopulation LD is defined as $D_M \equiv \bar{f}_{AB} - \bar{f}_A \bar{f}_B$. A more informative, but mathematically equivalent, description of metapopulation LD, however, is to define it in terms of the population variables as

$$D_M \equiv \bar{D} + \text{cov}(f_A, f_B), \tag{5}$$

that is, $D_M$ is the sum of the average population LD, $\bar{D}$, and the spatial covariance between the frequencies of resistance to drugs $A$ and $B$. *Equation (5)* shows that even if there is no population LD, that is, $D^x = 0$ and so $\bar{D} = 0$, there may still be metapopulation LD; likewise, there may be population LD, $D^x \neq 0$, but no metapopulation LD, $D_M = 0$ (*Nei and Li, 1973*; *Ohta, 1982a*; *Feldman and Christiansen, 1974*).

With this in mind, the change in frequency of infections resistant to drug $A$ (*mutatis mutandis* drug $B$) can be written

$$\frac{d\bar{f}_A}{dt} = \overbrace{\bar{s}_A \bar{f}_A (1 - \bar{f}_A)}^{\text{direct selection}} + \overbrace{\bar{s}_B D_M}^{\text{indirect selection}} + \overbrace{\bar{s}_E \bar{f}_A (1 - \bar{f}_A) \frac{\bar{f}_{AB}}{\bar{f}_A}}^{\text{epistasis}}$$

$$+ \underbrace{(\bar{\mu}_A + \bar{\rho}_A)(1 - \bar{f}_A) - (\bar{\mu}_a + \bar{\rho}_a)\bar{f}_A}_{\text{mutation and recombination}} + \underbrace{\text{cov}(r, f_A)}_{\text{heterogeneity in 'baseline' growth}} + \underbrace{\bar{f}_B \text{cov}\left(s_B, \frac{f_{AB}}{f_B}\right)}_{\text{heterogeneity in indirect selection}} . \tag{6}$$

The first four terms in *Equation (6)* are the metapopulation-level analogues of the first four terms in *Equation (3)* and so share the same interpretation. The last two terms, however, arise due to spatial heterogeneity in 'baseline' growth and selection and so are the consequence of population structure. As these terms are zero in the absence of spatial heterogeneities, they will be our focus here.

First, spatial heterogeneity arises through differences in the 'baseline' per-capita growth (i.e. $r^x \neq r^y$) coupled with differences in the frequencies of drug $A$ resistant infections (i.e. $f_A^x \neq f_A^y$). This is the spatial covariance between 'baseline' per-capita growth and the frequency of drug $A$ resistant infections, $\text{cov}(r, f_A)$. In particular, more productive populations (larger $r^x$) will have a disproportionate effect on the change in drug $A$ resistance. For example, if more productive populations also have a greater frequency of drug $A$ resistance, then heterogeneity increases the population frequency of drug $A$ resistance. Heterogeneity in baseline growth could arise through a variety of mechanisms, such as availability of susceptible hosts, treatment rates differences, or pathogen traits (e.g. transmissibility and duration of carriage).

Second, spatial heterogeneity arises through differences in indirect selection for resistance to drug $B$ (i.e. $s_B^x \neq s_B^y$) coupled with differences in the probability that drug $B$ resistant infections are also doubly-resistant (i.e. $f_{AB}^x / f_B^x \neq f_{AB}^y / f_B^y$). This is the spatial covariance between selection on resistance to drug $B$ and the conditional probability that a drug $B$ resistant infection is doubly-resistant, $\text{cov}(s_B, f_{AB}/f_B)$. In particular, populations experiencing greater selection for resistance to one drug will have a disproportionate effect on the change in frequency of infections resistant to the other drug, whenever populations differ in frequency of doubly-resistant infections. As an example, if populations experiencing stronger selection for drug $B$ resistance also have a greater probability of drug $B$-resistant infections being doubly-resistant, heterogeneity in indirect selection increases the frequency of drug $A$ resistance in the metapopulation.

Next, the dynamics of metapopulation LD can be written as

$$\frac{\mathrm{d}D_{\mathrm{M}}}{\mathrm{d}t} = \overbrace{(\bar{s}_A - \bar{s} + \bar{s}_B - \bar{s})D_{\mathrm{M}}}^{\text{selection}} - \overbrace{(\bar{\mu} + \bar{\rho})D_{\mathrm{M}}}^{\text{mutation and recombination}} + \overbrace{\bar{s}_E \bar{f}_{ab} \bar{f}_{AB}}^{\text{epistasis}}$$

$$+ \underbrace{\mathrm{cov}(r,D) + \mathrm{coskew}(r,f_A,f_B)}_{\text{heterogeneity in 'baseline' growth}} + \sum_{d \in \{A,B\}} \underbrace{(1 - \bar{f}_d)\bar{f}_d \, \mathrm{cov}\left(s_d, \frac{f_{AB}}{f_d}\right)}_{\text{heterogeneity in resistance selection}} \quad , \tag{7}$$

where $\mathrm{coskew}(r,f_A,f_B)$ is the spatial coskewness between $r$, $f_A$, and $f_B$ and we have assumed population differences in mutation and recombination are negligible (see Materials and methods 'Metapopulation LD and MDR'). The first three terms in *Equation (7)* are the metapopulation level analogues of the first three terms of *Equation (4)* and so share the same interpretation. The last two terms, however, arise due to spatial heterogeneity in 'baseline' growth and selection and so will be our focus here.

First, spatial heterogeneity arises through spatial differences in the 'baseline' per-capita growth (i.e. $r^x \neq r^y$) coupled with spatial heterogeneities in LD (i.e. $D^x \neq D^y$) or resistance frequencies (the coskewness term). The logic of the first term is clear: when population LD differs, more productive populations will disproportionately contribute to metapopulation LD. For the second term, when populations covary in frequency of resistance to drug $A$ and $B$, more productive populations will disproportionately contribute to the covariance, $\mathrm{cov}(f_A,f_B)$ and so disproportionately contribute to metapopulation LD (through the second term in *Equation (5)*).

Second, spatial heterogeneity arises through differences in selection for resistance ($s_d^x \neq s_d^y$) coupled with differences in the proportion of drug $d$ resistant infections that are doubly-resistant ($f_{AB}^x/f_d^x \neq f_{AB}^y/f_d^y$). The logic here is that populations experiencing stronger selection for resistance are more likely to see an increase in resistant infections. If this increase occurs disproportionately in doubly-resistant infections, then from *Equation (1)* metapopulation LD will increase, whereas if this increase occurs disproportionately in singly-resistant infections, metapopulation LD will decrease. The magnitude of this effect is scaled by $\bar{f}_d(1 - \bar{f}_d)$ since selection cannot operate without genetic variation. In the absence of population LD, then $f_{AB}^x/f_A^x = f_B^x$ and $f_{AB}^x/f_B^x = f_A^x$, and so if populations experiencing stronger selection for resistance to one drug also have a greater frequency of infections resistant to the other drug, metapopulation LD will increase. This could occur if, for example, some populations experience greater treatment rates.

As a final note, observe that in contrast to *Equation (4)*, in *Equation (7)* the per-capita migration rates $m^{y \rightarrow x}$ are nowhere to be found. The reason for this is intuitive: as migration does not affect the total density of infecteds, nor the resistance status of an infection, it will not change the quantities $\bar{f}_{AB}$, $\bar{f}_A$, or $\bar{f}_B$, and so cannot change metapopulation LD. As a consequence, migration only affects metapopulation LD indirectly by reducing differences in infection frequency between populations, thereby dampening the magnitude (and hence the effect) of $\mathrm{cov}(r,D)$, $\mathrm{cov}(r,f_d)$, and $\mathrm{cov}(s_\ell, f_{AB}/f_d)$ in *Equation (7)*. It follows that, all else being equal, the magnitude of $D_{\mathrm{M}}$ is a decreasing function of the per-capita migration rate, and so is maximized when migration is infrequent (*Figure 2*).

## Modeling the dynamics of LD: why bother?

To this point, we have focused upon developing the LD perspective to provide a conceptual understanding of MDR evolution in structured populations. However, framing the LD perspective in terms of general quantities has meant this conceptual understanding is somewhat abstract. What we now wish to demonstrate, through the consideration of three scenarios, is how the LD perspective can be used to tackle practical problems. In the first scenario, we show how the LD perspective provides additional insight into a recent paper on the effect of spatial structure on equilibrium patterns of MDR. In the second scenario, we show how the LD perspective allows for an understanding of transient dynamics, and we apply this understanding to patterns of MDR observed in *Streptococcus pneumoniae*. In the third scenario, we show how the LD perspective generates practical insight into designing drug prescription strategies across populations, with a focus upon a hospital-community setting.

## LD perspective explains equilibrium patterns of MDR

Understanding the patterns of MDR in structured populations was first tackled in an important paper by *Lehtinen et al., 2019*. The paper by *Lehtinen et al., 2019* (see also *Jacopin et al., 2020*) focused upon MDR evolution in a metapopulation consisting of independent host populations (so migration is restricted, $m^{x \to y} \approx 0$). For example, each population could represent a different *Streptococcus pneumoniae* serotype maintained by serotype-specific host immunity (*Henriques-Normark and Tuomanen, 2013*; *Cobey and Lipsitch, 2012*; *Lehtinen et al., 2017*). *Lehtinen et al., 2019* found that at equilibrium, population differences could lead to MDR over-representation ($D_M > 0$), and that populations with a longer duration of pathogen carriage were more likely to exhibit MDR, a result they attributed to an increased likelihood of antibiotic exposure per carriage episode. Here, we show how employing the LD perspective: (i) reveals the evolutionary logic behind what populations differences can maintain metapopulation LD at equilibrium and (ii) using these insights allows us to build upon the results of *Lehtinen et al., 2019* to understand how epidemiological factors other than duration of carriage can play an important role. For simplicity, we will assume that costs are additive (see *Box 1*), and so there is no epistasis (i.e. $s_E^x = 0$), but as this differs from *Lehtinen et al., 2019* who use multiplicative costs, we discuss this assumption in more depth in Materials and methods 'Equilibrium analysis of metapopulation consisting of independent populations'.

To maintain metapopulation LD at equilibrium, there needs to be at minimum some mechanism maintaining metapopulation resistance diversity, otherwise $D_M = 0$. There are variety of ways in which this could occur (*Lipsitch et al., 2009*; *Colijn et al., 2010*; *Davies et al., 2019*; *Lehtinen et al., 2017*; *Jacopin et al., 2020*; *Krieger et al., 2020*), but *Lehtinen et al., 2017*, *Lehtinen et al., 2019* assume it is due to some variation among populations in the conditions favoring resistance evolution. This mechanism maintains diversity at the scale of the metapopulation but leads to the fixation or the extinction of drug resistance locally. Thus $D^x = 0$, and it follows from *Equation (5)* that $D_M = \mathrm{cov}(f_A, f_B)$. Therefore, in order for metapopulation LD to exist, $f_A$ and $f_B$ must covary across populations. Specifically, whenever $f_A^x$ and $f_B^x$ (or their dynamical equations, *Equation 3*), are uncorrelated, the metapopulation will be in linkage equilibrium. From *Equation (3)* we see that if the additive selection coefficients, $s_A^x$ and $s_B^x$, are uncorrelated, then so too are the dynamics of $f_A^x$ and $f_B^x$, and so $\mathrm{cov}(f_A, f_B) = 0$. Hence only when population differences generate correlations between the selection coefficients will they generate LD.

Using this insight, why are populations with a longer duration of carriage associated with MDR (*Lehtinen et al., 2019*)? And should we expect associations between MDR and any other population attributes? Our primary focus is whether (and how) the selection coefficients are correlated. Letting $\Delta z_k^x$ be the contribution of trait $z$ to the additive selection coefficient for resistance to drug $d$ in population $x$ (e.g. $\Delta \beta_A^x = \beta_{Ab}^x - \beta_{ab}^x$), then it is straightforward to compute (see Materials and methods 'Equilibrium analysis of metapopulation consisting of independent populations'),

$$
\begin{aligned}
s_A^x &= \Delta \beta_A^x S^x - \Delta \alpha_A^x + \tau_A^x, \\
s_B^x &= \Delta \beta_B^x S^x - \Delta \alpha_B^x + \tau_B^x.
\end{aligned}
\tag{8}
$$

where we have used slightly different notation from *Lehtinen et al., 2019*. Now, consider a scenario in which both the treatment rates and the parameters controlling the (additive) costs of resistance are uncorrelated (i.e. $\Delta \beta_d^x = \Delta \beta_d$, $\Delta \alpha_d^x = \Delta \alpha_d$ and $\tau_d^x = \tau_d$); this is one of the scenarios presented in Figure 4 of *Lehtinen et al., 2019*, with the key difference that they considered 'multiplicative' rather than 'additive' costs. From *Equation (8)*, the only remaining source of correlation is susceptible density, $S^x$, which plays a role whenever there are explicit transmission costs, $\Delta \beta_d < 0$. Although *Equation (8)* always holds, in keeping with *Lehtinen et al., 2019* if we focus upon the equilibrium case, $S^x$ will be determined by pathogen traits such as transmission and duration of carriage, such that 'fitter' populations (i.e. those in which pathogens are more transmissible or have longer duration of carriage) will more substantially deplete susceptibles. By reducing $S^x$, 'fitter' populations lower the transmission costs for resistance to either drug, and so double-resistance is more likely to be selectively advantageous, even when treatment rates are uncorrelated. In turn, this over-representation of doubly-resistant infections will generate metapopulation LD.

Thus, when costs are 'additive' (*Box 1*), although variation in duration of carriage can lead to MDR evolution and LD through its effect upon susceptible density (*Figure 3a*), it is neither necessary (the same pattern can be produced by variation in transmissibility; *Figure 3b*) nor sufficient (variation

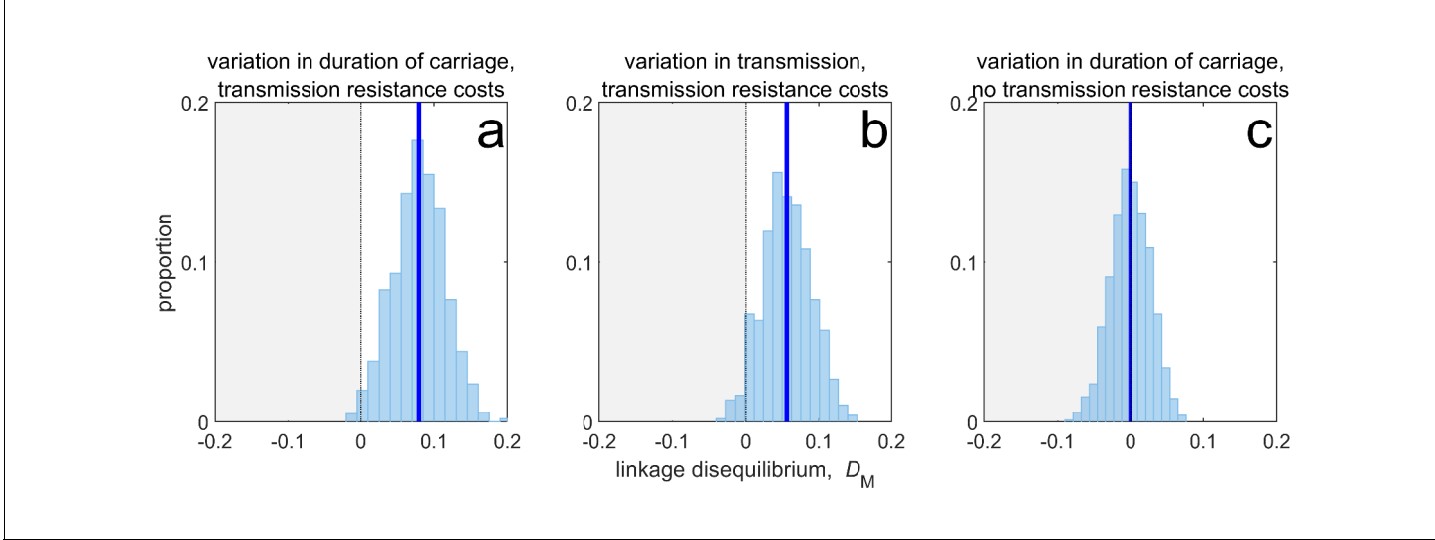

**Figure 3.** Duration of carriage is one of many potential explanations for MDR over-representation at equilibrium. When costs are additive and there is no epistasis, variation in duration of carriage across independent populations can lead to linkage disequilibrium (subplot a), but it is neither necessary (**b**), nor sufficient (**c**). We simulate 1000 populations (blue bars), each consisting of 20 independent populations in which treatment rates for each population are randomly chosen to be either $\tau_{\max} = 0.075$ or $\tau_{\min} = 0.025$ with equal probability while simultaneously satisfying $\mathrm{cov}(\tau_A, \tau_B) = 0$. The solid blue line is the mean LD across the simulations for each scenario. In subplot a, duration of carriage varies across populations and there are transmission resistance costs; in subplot b, transmission varies and there are transmission resistance costs; while in subplot c, duration of carriage varies and there are no transmission costs. These simulations diverge slightly from those of *Lehtinen et al., 2019* in that their model always includes epistasis (see Materials and methods 'Equilibrium analysis of metapopulation consisting of independent populations'), whereas here we only consider non-epistatic scenarios.

in duration of carriage has no effect without explicit transmission costs, *Figure 3c*). More broadly, if there are more than two drugs, then provided that there are explicit transmission costs for resistance to each drug, susceptible density will generate a correlation between all the selection coefficients, which in turn will yield the pattern of 'nestedness' observed by *Lehtinen et al., 2019*. What is critical for this effect to be prominent, however, is (i) the existence of population differences in susceptible density, and (ii) the costs of resistance (i.e. $\Delta\beta_d$), are large enough so as to ensure a strong correlation amongst selection coefficients.

## LD perspective explains transient patterns of MDR

The predictions of *Lehtinen et al., 2019* were used to explain the patterns of MDR observed in surveillance data. One of these data sets was a surveillance study that documented both the serotype as well as antibiotic resistance to a number of different drugs in *S. pneumoniae* infections sampled in Maela, Northern Thailand (*Turner et al., 2012*; *Lehtinen et al., 2019*). Although the prediction of positive (metapopulation) LD was met for most drug combinations (*Lehtinen et al., 2019*), inspection of the data set reveals significant serotype LD (*Figure 4*). This is notable because, as we have detailed above, at equilibrium the simplest version of the model used in the previous section will result in each serotype being in linkage equilibrium, $D^x = 0$. How can we reconcile these conflicting observations? Although there are various possible explanations (e.g. an additional mechanism capable of maintaining diversity within-serotype), here we focus upon relaxing the assumption that the metapopulation is at equilibrium. That is, we are interested in whether long-term transient dynamics unfolding over months and years could plausibly suggest an alternative explanation for the observed serotype LD.

To do so, consider a metapopulation consisting of independent serotypes, differing in their transmissibility and duration of carriage (as in the model of *Lehtinen et al., 2017*; *Lehtinen et al., 2019*). Assume that there is no epistasis and that the additive selection coefficients take the form of *Equation (8)*, where the parameters $\Delta\beta_d^x$, $\Delta\alpha_d^x$ and $\tau_d^x$ do not depend upon serotype $x$ (Materials and methods 'Transient dynamics and MDR in streptococcus pneumoniae'). Suppose that initially the

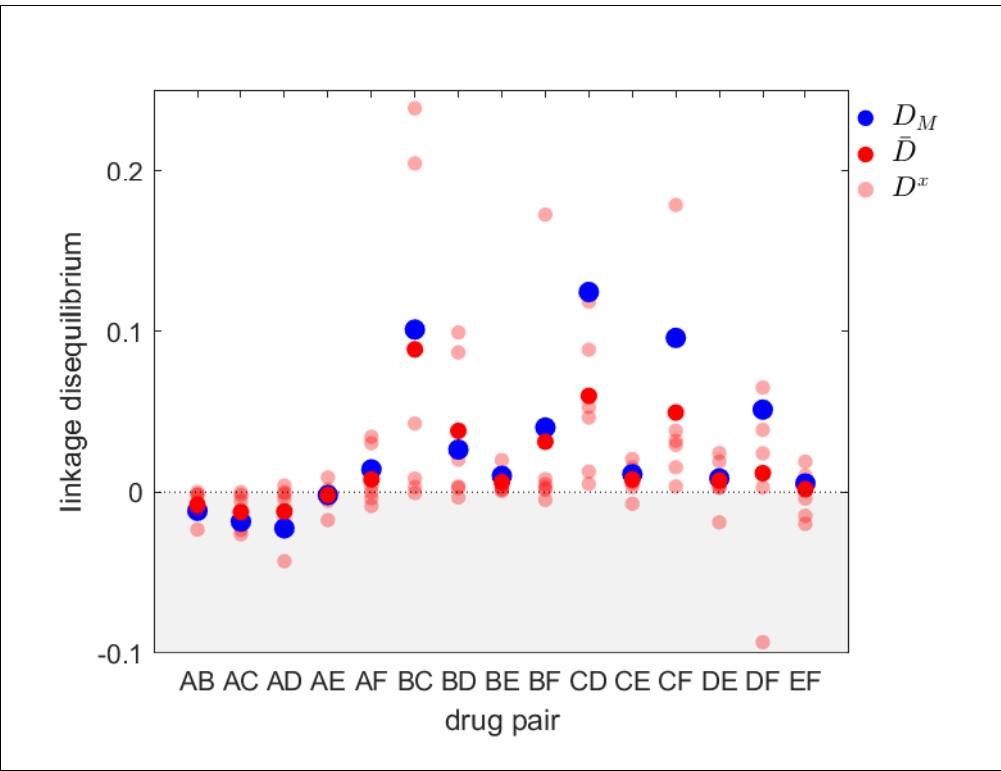

**Figure 4.** Linkage disequilibrium for different drug pairs in *Streptococcus pneumoniae*. Data is from the Maela surveillance data set of *Lehtinen et al., 2019*; *Turner et al., 2012*. The light red circles are the observed serotype LD, $D^x$, the dark red circles are the average LD across serotypes, $\bar{D}$, while the blue circles are the metapopulation LD, $D_{\mathrm{M}}$. We have restricted the data to serotypes involving 100 or more samples (serotypes 14, 6A/C, 6B, 15B/C, 19F, 23F). The drugs considered are: A = chloramphenicol, B = clindamycin, C = erythromycin, D = penicillin, E = sulphatrimethoprim, and F = tetracycline.

metapopulation is treated exclusively with drug $A$ at sufficiently high rates such that resistance to drug $A$ goes to fixation in each serotype, that is, $\bar{f}_A \to 1$ and $\bar{f}_B \to 0$, and so $D_{\mathrm{M}} = 0$. Now suppose at time $t = 1000$ months that drug $B$ is 'discovered' and subsequently prescribed at a high rate in the metapopulation, while owing to its reduced efficacy, prescription of drug $A$ is reduced. Although the treatment rates do not vary by serotype, serotype differences in transmissibility and duration of carriage mean that the changes to treatment rates will differentially affect serotype density, which in turn will differentially affect the serotype-specific availability of susceptible hosts, $S^x$. Since the serotype-specific selection coefficients, $s_A^x$ and $s_B^x$, and baseline per-capita growth, $r^x$, directly depend upon $S^x$, the variation in $S^x$ introduces heterogeneity in *Equation (7)*, which in turn generates metapopulation LD. Because the selection coefficients are positively correlated (due to the shared dependence upon $S^x$), the metapopulation LD generated will be positive, that is, $D_{\mathrm{M}} > 0$ (*Figure 5a,d*). From *Equation (7)*, once metapopulation LD is generated, it will be amplified by directional selection (first term of *Equation 7*) which is initially positive since resistance to drug $B$ is favored; this leads to a rapid build up of $D_{\mathrm{M}}$ (*Figure 5a,d*). However, this initial increase in $D_{\mathrm{M}}$ is transient; for this particular choice of parameter values, at equilibrium $D_{\mathrm{M}} \to 0$. Crucially, however, the changes to $D_{\mathrm{M}}$ can unfold over a very long time (here the time units are months), such that surveillance data would detect little change in the metapopulation dynamics and so suggest a population roughly in equilibrium.

Although this scenario can lead to considerable (transient) metapopulation LD, there is still nothing generating serotype LD. Our analysis of *Equation (4)* revealed two possible (deterministic) mechanisms capable of generating population (serotype) LD. First, migration between populations can lead to metapopulation LD spilling over into population LD. In this example, 'migration' between

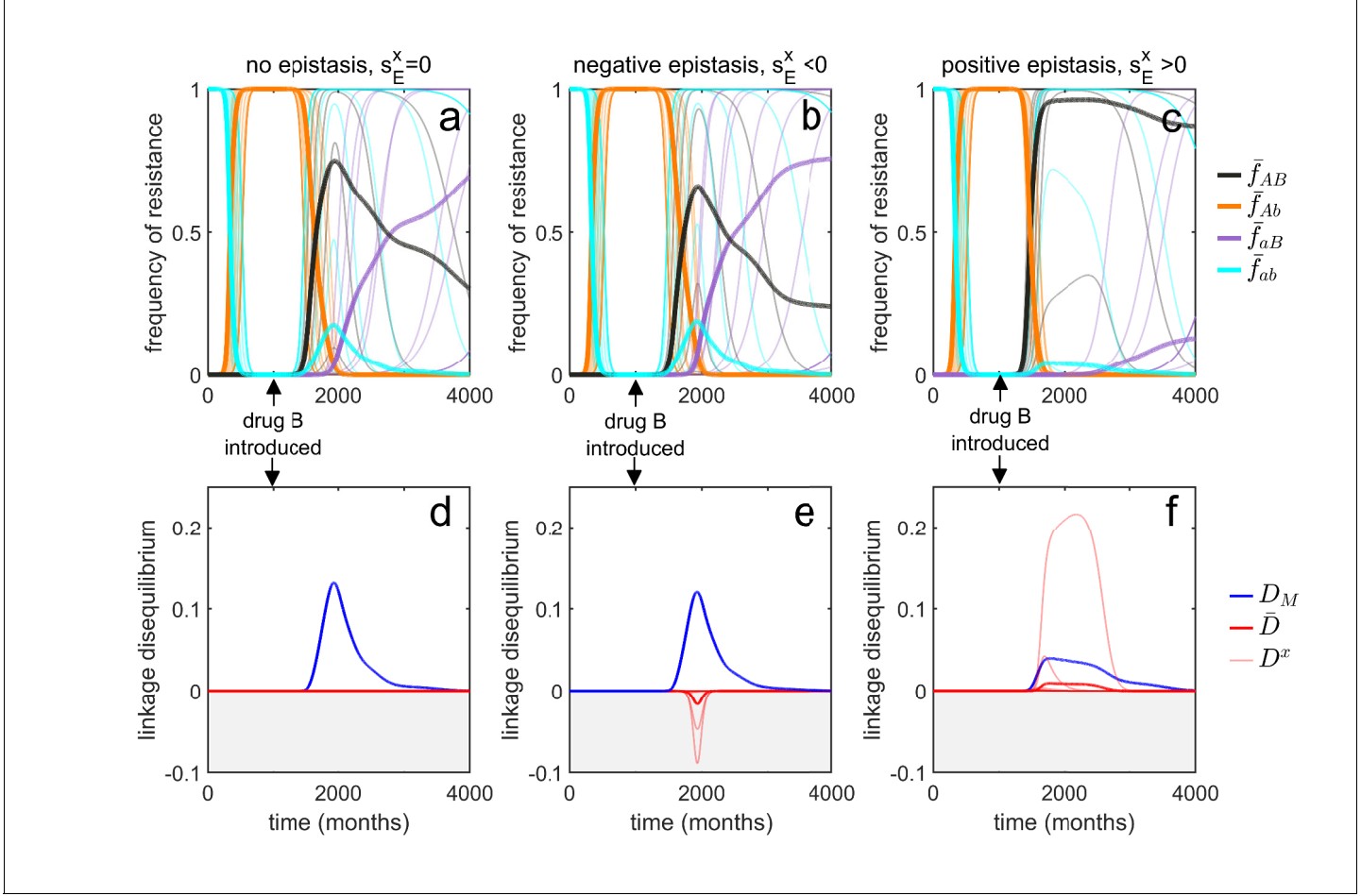

**Figure 5.** Transient dynamics coupled with epistasis can explain patterns of serotype LD in *Streptococcus pneumoniae*. In all simulations, serotypes differ based upon duration of carriage and transmissibility. At $t = 0$, the pathogen is sensitive to both drugs; however, as hosts are initially treated with drug $A$ at a rate of $\tau_A = 0.12$ per month, resistance to drug $A$ emerges and fixes in all serotypes. At $t = 1000$ (months), drug $B$ is introduced, and drug $A$ prescription reduced, $(\tau_A, \tau_B) = (0.07, 0.1)$ (note that the drugs are never prescribed in combination, $\tau_{AB} = 0$). In the first column, there is no epistasis, thus although metapopulation LD builds up, serotype LD does not. In the second column, there is negative epistasis, which generates negative serotype LD. In the third column, there is positive epistasis which produces positive serotype LD. The thin lines denote the within-serotype dynamics, while the thick lines denote the metapopulation dynamics. In all cases, at equilibrium both the serotypes and the metapopulation will be in linkage equilibrium, however, transient LD can occur on sufficiently long timescales so as to appear permanent (see Materials and methods 'Transient dynamics and MDR in streptococcus pneumoniae' for more details).

serotypes would correspond to serotype 'switching' (*Croucher et al., 2015*), whereby infections exchange serotypes through recombination. However, this is an unlikely explanation as the rate of serotype switching would have to be unrealistically large for serotype LD to be substantially altered. The second term in *Equation (4)* capable of generating serotype LD is epistasis, which will generate same sign serotype LD (*Felsenstein, 1965*; *Lewontin, 1964*; *Lewontin and Kojima, 1960*). Indeed, in the model considered, negative epistasis, $s_E^x < 0$, generates transient negative serotype LD (*Figure 5b,e*), while positive epistasis generates transient positive serotype LD (*Figure 5c,f*; Materials and methods 'Transient dynamics and MDR in streptococcus pneumoniae'). Notably, although negative epistasis produces negative serotype LD (and so $\bar{D} < 0$), at the scale of the metapopulation this effect is swamped by the positive covariance in frequency of resistance and so metapopulation LD is positive, $D_M > 0$. (*Figure 5e*).

Thus, transient dynamics coupled with epistasis could provide a potential explanation for the significant within-serotype LD observed in *S. pneumoniae* (*Figure 4*). More generally, the potential complexity of competing selective pressures associated with multilocus dynamics can lead to

prolonged, but transient, polymorphisms and LD, and so surveillance data showing limited temporal change in resistance frequency should be treated cautiously and not assumed to be due to a stable equilibrium.

## LD perspective helps identify drug prescription strategies limiting the evolution of MDR

Understanding the evolutionary consequences of different antibiotic prescription strategies across populations can have practical relevance for public health. The populations of interest could correspond to physically distinct groups such as a hospital and its broader community, or different geographical regions (e.g. countries). From a public health perspective, when considering different prescription strategies, a variety of factors must be considered, but in general the goal is to successfully treat as many people as possible, thereby reducing the total burden (*Bonhoeffer et al., 1997*; *Abel zur Wiesch et al., 2014*). In this circumstance, the LD in the metapopulation and/or populations can provide important information about the likelihood of treatment success. In particular, for a given population frequency of drug $A$ and drug $B$ resistance, negative LD (MDR under-representation) increases the likelihood that if treatment with one drug fails (due to resistance), treatment with the other drug will succeed. On the other hand, positive LD (MDR over-representation) increases the likelihood of treatment failure, since a greater proportion of resistant infections are doubly-resistant and so cannot be successfully treated with either drug. *Equations (4) and (7)* show that to generate negative LD, drugs should be deployed in a population specific fashion, that is, drug $A$ should be restricted to some populations and drug $B$ restricted to the remaining populations (see also *Lehtinen et al., 2019*; *Day and Gandon, 2012*; *Jacopin et al., 2020*). Doing so will create a negative covariance in selection, such that resistance to drug $A$ (resp. drug $B$) will be favored in some populations and disfavored in the others. This negative covariance in selection will give rise to negative LD and MDR under-representation (*Figure 2*).

As an application of this principle, consider two populations connected by migration, corresponding to a 'community' and a much smaller 'hospital'. Drug prescription occurs at a fixed (total) rate in each population, while the prescription rate is much higher in the hospital (see Materials and methods 'Contrasting drug prescription strategies in a hospital-community setting'). Consider three antibiotic prescription strategies: (i) drugs can be randomly prescribed to individuals (*mixing*); (ii) drugs can be prescribed exclusively in *combination*; or (iii) prescription of drug $A$ and $B$ can be asynchronously rotated between the hospital and community, that is, if the hospital uses drug $A$ then the community uses drug $B$, and vice versa (*cycling*). As both drugs are prescribed at higher rates in the hospital than the community, both mixing and combination generate a positive covariance in selection across populations, producing positive LD and MDR over-representation (see *Equation (4)*; *Figure 6*). Thus, over the short- and long-term, mixing and combination produce similar results: doubly-resistant infections are favored, while singly-resistant infections are disfavored (*Figure 6*). Now consider cycling. When drugs are rotated rapidly between populations, infections in either population are likely to be exposed to both drugs. Because prescription rates are higher in the hospital, this effectively creates a positive covariance in selection (i.e. cycling behaves like mixing) and so when resistance emerges, infections tend to be doubly-resistant (MDR over-representation). When drugs are rotated less frequently, infections are more likely to be exposed to a single drug, creating a negative covariance in selection across populations. In this circumstance, although single resistance can emerge at lower treatment rates then when rotations are more frequent, the negative LD produced by the negative covariance in selection inhibits the emergence of double-resistance (MDR under-representation; *Figure 6*).

These results emphasize an important trade-off: delaying the evolution of MDR (e.g. by decreasing time between rotations) promotes the evolution of single drug resistance, whereas delaying the evolution of single drug resistance promotes the evolution of MDR. This is logical: when we maintain a constant treatment rate per individual, decreasing selection for the 'generalist' strategy (MDR) necessarily increases selection for the 'specialist' strategy (single drug resistance) (*Wilson and Yoshimura, 1994*). Thus, cycling can either be the best, or worst, option for single drug resistance (*Beardmore et al., 2017*), but critically, this has concomitant effects for MDR (see also *Figure 6*). Indeed, mixing, combination and cycling have been exhaustively compared in the context of single drug resistance (e.g. *Bonhoeffer et al., 1997*; *Lipsitch et al., 2000*; *Bergstrom et al., 2004*;

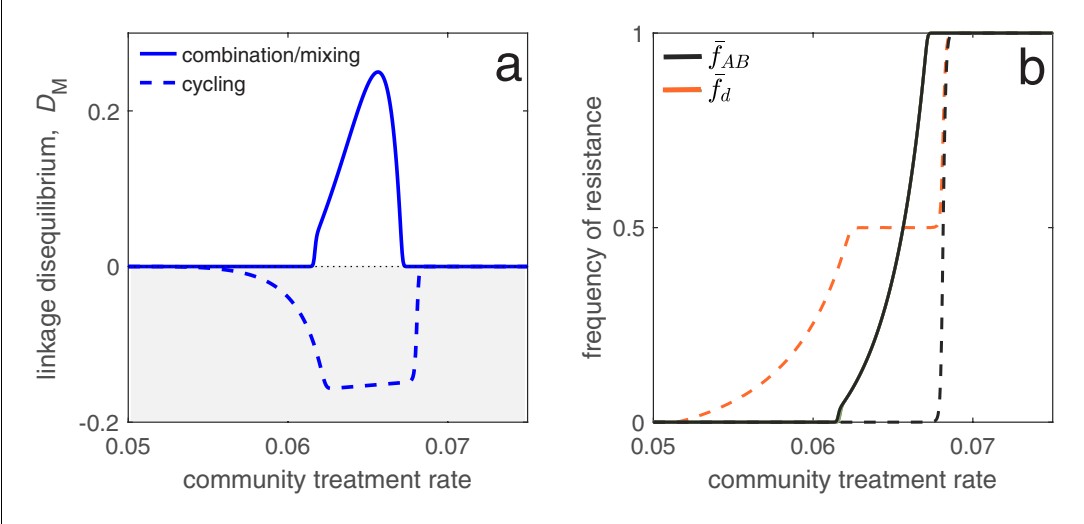

**Figure 6.** Different antibiotic prescription strategies generate different patterns of LD at equilibrium. Here, we focus upon a population divided into a community and a hospital. Individuals enter the hospital at a fixed rate and spend a fifth of the time in the hospital that it takes to naturally clear a sensitive infection. The hospital/community size split corresponds to 20 beds per 1000 people, while individuals in the hospital receive antibiotics at 15x the rate they do in the community. We integrate system (2) until equilibrium is reached; the final state of the system is what is shown. For cycling, we compute the average state over the last two rotations (i.e., over the last period, $T$; in this case $T = 100$). In panel a, we show the metapopulation LD, $D_{\mathrm{M}}$ for the three treatment scenarios (combination, mixing, cycling). Combination and mixing generate identical LD in this example. In panel b we show the frequency of infections in the metapopulation resistant to drug $d$, $\bar{f}_d$ (for our choice of parameters, $\bar{f}_A = \bar{f}_B$; Materials and methods 'Contrasting drug prescription strategies in a hospital-community setting'), and doubly-resistant, $\bar{f}_{AB}$, for each scenario. Note that for mixing and combination treatments (solid curves), $\bar{f}_A = \bar{f}_B = \bar{f}_{AB}$, whereas cycling (dashed curves) leads to singly-resistant infections at low treatment rates (see Materials and methods 'Contrasting drug prescription strategies in a hospital-community setting').

*Beardmore and Pena-Miller, 2010*; *Beardmore et al., 2017*; *Abel zur Wiesch et al., 2014*; *Tepekule et al., 2017*); yet these studies largely ignored the consequences for MDR evolution. Our analysis suggests controlling for single drug resistance will have important consequences for MDR, and so it should not be considered in isolation. More generally, whether it is optimal to either delay single drug resistance or prevent MDR will depend upon what metric is used to evaluate what constitutes a 'success' or 'failure'.

## Discussion

The evolution of multidrug-resistant pathogens is a pressing health concern and is a topic which is increasingly gaining attention from evolutionary biologists and mathematical modellers alike. However, the typical process in studying the problem of MDR is to introduce a model of the form of (2), and then either proceed to a numerical analysis of these equations or simplify the model further by neglecting the dynamics of double resistant infections (*Bergstrom et al., 2004*; *Bonhoeffer et al., 1997*; *Beardmore et al., 2017*). This is because models of MDR evolution rapidly become intractable, a problem which is particularly acute when incorporating aspects of population structure. Here, we have argued that a more insightful and simplifying approach is the 'linkage disequilibrium perspective': after specifying the model of interest, as in (2), it is desirable to transform the model into the form of *Equations (3), (4), (6), and (7)*, which brings to the forefront the role played by linkage disequilibrium for MDR evolution in structured populations. The LD perspective is particularly useful for analyzing and understanding transient evolutionary dynamics (*Figure 5*), which cannot be understood by, for example, invasion analysis.

Our analysis emphasizes that metapopulation structure alone can generate and maintain LD (and so MDR), even in the absence of epistasis (*Nei and Li, 1973*; *Ohta, 1982a*; *Li and Nei, 1974*; *Slatkin, 1975*). Since in natural populations metapopulation structure is often hidden (e.g. *Rosen et al., 2015*), patterns of MDR should not be assumed to be due to epistasis, even if no structure is readily apparent. Moreover, caution must be taken when measuring LD (and MDR) at a particular scale, as

doing so can lead to erroneous conclusions: even if the metapopulation is in linkage equilibrium, $D_{\mathrm{M}} = 0$, the populations need not be, $D^x \neq 0$, and vice versa (*Equation (5)*), while in more extreme cases, population and metapopulation LD can be of opposite sign (*Figure 5b,e*). These are not merely esoteric points; the presence or absence of LD (and MDR), and its source (epistasis or metapopulation structure) is critically important. For example, when MDR is due to metapopulation structure rather than epistasis, prescribing drugs across different populations so as to create a negative covariance in selection can reduce the prevalence of MDR (*Figure 6*; *Day and Gandon, 2012*; *Jacopin et al., 2020*; *Lehtinen et al., 2019*), while distinguishing between population and metapopulation LD can provide additional insight toward evaluating hypotheses (*Figures 4* and *5*).

Our analysis assumed that the evolutionary dynamics were deterministic, thus neglecting the influence of stochasticity. However, it is widely appreciated in population genetics that stochasticity can play an important role in multilocus dynamics. For example, LD can be generated through genetic drift (*Hill and Robertson, 1966*; *Barton, 1995*; *Lenormand and Otto, 2000*; *Otto and Barton, 2001*; *Keightley and Otto, 2006*; *Martin et al., 2006*), which in turn can interfere with the strength of selection (*Hill and Robertson, 1966*; *Neher and Shraiman, 2011*; *Slatkin, 2008*). Similarly, the (random) genetic background a rare mutation finds itself upon is critically important for its success (*Kouyos et al., 2006*; *Gillespie, 2000*; *Neher, 2013*), and in finite populations this alone can generate LD. However, little has been done to relate these results to evolutionary epidemiology, or to understand how epidemiological feedbacks can influence their predictions. The little work to date has relied upon complex simulations (e.g. *Althaus and Bonhoeffer, 2005*; *Kouyos et al., 2009*), which necessarily sacrifice general insight for specificity. Thus, the role of stochasticity in the evolution of MDR remains an area in which further investigation is warranted.

Understanding the evolution of MDR is a research topic of pressing concern. Here, we have argued that using the linkage disequilibrium perspective leaves us better equipped to determine what factors are responsible for generating MDR, and their generality. Moreover, taking such an approach leads to a more straightforward comparison with existing models and results.

## Materials and methods

Here, we provide more comprehensive details on the analysis presented in the main text. We start by deriving the general epidemiological model for the dynamics of the different strains which are characterised by their multilocus genotype (Materials and methods 'Model derivation'). We then convert this model into an equivalent system which tracks the dynamics of allele frequencies at each locus, and the LD at the population level (Materials and methods 'Population LD and MDR'), before considering the set of equations for the dynamics of allele frequencies at each locus and the LD at the metapopulation level (Materials and methods 'Metapopulation LD and MDR').

We conclude by providing a detailed mathematical analysis of the three examples presented in the main text: (1) using the LD perspective to explain equilibrium patterns of MDR (Materials and methods 'Equilibrium analysis of metapopulation consisting of independent populations'); (2) using the LD perspective to explain transient patterns of MDR (Materials and methods 'Transient dynamics and MDR in streptococcus pneumoniae'); and (3) applying the LD perspective to identify drug prescription strategies limiting MDR evolution (Materials and methods 'Contrasting drug prescription strategies in a hospital-community setting').

### Model derivation

Our focus is on an asymptomatically carried bacteria species in a metapopulation consisting of $N$ populations. Focus upon an arbitrarily chosen population $x$. Let $S^x$ and $I_{ij}^x$ denote the density of susceptible hosts and $ij$-infections, respectively, at time $t$, where $i$ indicates if the infection is resistant ($i = A$) or not ($i = a$) to drug $A$ and $j$ indicates if the infection is resistant ($j = B$) or not ($j = b$) to drug $B$. Susceptible hosts contract $ij$-infections at a per-capita rate $\beta_{ij}^x I_{ij}^x$, where $\beta_{ij}^x$ is a rate constant, while $ij$-infections are naturally cleared at a per-capita rate $\alpha_{ij}^x$. Hosts in population $x$ are treated with antibiotics $A$, $B$, or both in combination, at per-capita rates $\tau_A^x$, $\tau_B^x$, and $\tau_{AB}^x$, respectively. Hosts move from population $x$ to population $y$ at a per-capita rate $m^{x \to y}$.

The resistance profile of an infection changes through two processes. First, there may be de novo mutation, and so let $\mu_i^x$ be the per-capita rate at which an infection in population $x$ acquires allele $i$

through mutation. Second, a $ij$-infection may be super-infected by a $k\ell$-strain (*Day and Gandon, 2012*); in this circumstance recombination may occur. Specifically, $k\ell$-strains are transmitted to $ij$-infections at rate $\beta_{k\ell}^x I_{k\ell}^x I_{ij}^x$, whereupon with probability $\sigma$ super-infection occurs. In the event of super-infection, with probability $1 - \rho$, recombination does not occur, in which case with equal probability the $ij$-infection either remains unchanged or becomes a $k\ell$-infection. With probability $\rho$, recombination does occur, in which case with equal probability the $ij$-infection becomes either an $i\ell$- or $kj$-infection. Because our focus is upon the role of population structure, we do not allow for co-infection or within-host competitive differences based upon resistance profiles (e.g. *Davies et al., 2019*) but these are straightforward extensions. Moreover, at this stage, we do not make any further specification of the dynamics of uninfected hosts, be they susceptible or recovered, as doing so is not essential for a qualitative understanding of MDR evolution.

Rather than immediately writing down the set of differential equations corresponding to these epidemiological assumptions, we instead group the terms based upon the four biological processes that are occurring. In particular, the change in $I_{ij}^X$ can be written as the sum of:

1. The net change due to *mutation*, denoted $\phi\mu_{ij}^x$. As an example, focus upon the change in $Ab$-infections in population $x$ due to mutation, $\phi\mu_{Ab}^x$. These infections can increase through mutation in one of two ways: (i) $ab$-infections acquiring allele $A$ at rate $\mu_A^x I_{ab}^x$ or (ii) $AB$-infections acquiring allele $b$ at rate $\mu_b^x I_{AB}^x$. On the other hand, $I_{Ab}^x$ infections are lost due to mutation whenever they (i) acquire allele $a$ at a per-capita rate $\mu_a^x$, or (ii) acquire allele $B$ at a per-capita rate $\mu_B^x$. Combining this information gives the change in $Ab$-infections in population $x$ as

$$\phi\mu_{Ab}^x = \mu_A^x I_{ab}^x + \mu_b^x I_{AB}^x - (\mu_a^x + \mu_B^x)I_{Ab}^x, \tag{9}$$

   which is mathematically equivalent to

$$\phi\mu_{Ab}^x = \mu_A^x (I_{ab}^x + I_{Ab}^x) + \mu_b^x (I_{Ab}^x + I_{AB}^x) - \mu^x I_{Ab}^x, \tag{10}$$

   where $\mu^x \equiv \mu_a^x + \mu_A^x + \mu_b^x + \mu_B^x$ is the per-capita mutation rate in population $x$. The only difference between the two formulations is interpretation: *Equation (9)* shows only mutations which lead to a change in state, whereas *Equation (10)* shows all possible mutations, even those which do not. This is why the per-capita loss term, $\mu^x$, in (10) can be considered the total per-capita mutation rate in population $x$. More generally, we can write $\phi\mu_{ij}^x$ as

$$\phi\mu_{ij}^x \equiv \mu_i^x (I_{aj}^x + I_{Aj}^x) + \mu_j^x (I_{ib}^x + I_{iB}^x) - \mu^x I_{ij}^x. \tag{11}$$

2. The net change due to *recombination*, denoted $\phi\rho_{ij}^x$. Specifically, let $\rho_i^x$ be the per-capita rate at which infections gain allele $i$ through recombination. For example, consider $\rho_A^x$. In particular, $ij$-infections are challenged by strains carrying allele $A$ at rate $(\beta_{Ab}^x I_{Ab}^x + \beta_{AB}^x I_{AB}^x)I_{ij}^x$. With probability $\sigma$, a superinfection event occurs. Given an superinfection event, with probability $\rho$ recombination happens, in which case with probability $1/2$ the recombinant strain $Aj$ will replace the $ij$-infection. Thus

$$\rho_A^x = \rho\frac{\sigma}{2}(\beta_{Ab}^x I_{Ab}^x + \beta_{AB}^x I_{AB}^x), \tag{12}$$

   and $ij$-infections acquire allele $A$ in population $x$ at rate $\rho_A^x I_{ij}^x$. Therefore, the change in $ij$-infections in population $x$ due to recombination is

$$\phi\rho_{ij}^x \equiv \rho_i^x (I_{aj}^x + I_{Aj}^x) + \rho_j^x (I_{ib}^x + I_{iB}^x) - \rho^x I_{ij}^x \tag{13}$$

   where $\rho^x$ is the per-capita rate of recombination in population $x$, that is,

$$\rho^x \equiv \rho\sigma \sum_{k\ell} \beta_{k\ell}^x I_{k\ell}^x = \rho_a^x + \rho_A^x + \rho_b^x + \rho_B^x.$$

3. The net change due to host *migration* between populations,

$$-\sum_{y=1}^{N} m^{x \to y} I_{ij}^{x} + \sum_{y=1}^{N} m^{y \to x} I_{ij}^{y}. \tag{14}$$

4. The net change due to *per-capita growth*,

$$r_{ij}^{x} \equiv \beta_{ij}^{x} S^{x} - \alpha_{ij}^{x} - \mathbf{1}_a(i)\tau_A^x - \mathbf{1}_b(j)\tau_B^x - (1 - \mathbf{1}_A(i)\mathbf{1}_B(j))\,\tau_{AB}^x - (1-\rho)\frac{\sigma}{2}\sum_{k\ell}(\beta_{k\ell}^x - \beta_{ij}^x)I_{k\ell}^x,$$

where $\mathbf{1}_i(j)$ is an indicator variable and is equal to 1 if $i = j$ and 0 otherwise.

With these four processes in hand, the dynamics of infection densities are given by the system of $4N$ differential equations.

$$\frac{\mathrm{d}I_{ij}^{x}}{\mathrm{d}t} = \phi\mu_{ij}^{x} + \phi\rho_{ij}^{x} - \sum_{y=1}^{N}(m^{x \to y}I_{ij}^{x} - m^{y \to x}I_{ij}^{y}) + r_{ij}^{x}I_{ij}^{x}, \quad x=1,2,...,N,\ i \in \{a,A\},\ j \in \{b,B\}. \tag{15}$$

## Population LD and MDR

In what follows, we provide more details for the calculations of population LD and MDR. First, we define the following frequencies of infections in population $x$ as

$$f_A^x = \frac{\sum_j I_{Aj}^x}{I^x}, \quad f_B^x = \frac{\sum_i I_{iB}^x}{I^x}, \quad \text{and} \quad f_{ij}^x = \frac{I_{ij}^x}{I^x}, \tag{16}$$

where $I^x = \sum_{ij} I_{ij}^x$ is the total density of infections in population $x$. Using these definitions, the standard measure of linkage disequilibrium in population $x$ is

$$D^x = f_{AB}^x - f_A^x f_B^x, \tag{17}$$

which is mathematically equivalent to

$$D^x = f_{AB}^x f_{ab}^x - f_{Ab}^x f_{aB}^x. \tag{18}$$

The three dynamical equations of interest for studying MDR in population $x$ are

$$\frac{\mathrm{d}f_A^x}{\mathrm{d}t} = s_A^x f_A^x (1 - f_A^x) + s_B^x D^x + s_E^x f_A^x (1 - f_A^x)\frac{f_{AB}^x}{f_A^x} + (\mu_A^x + \rho_A^x)(1 - f_A^x) - (\mu_a^x + \rho_a^x)f_A^x - \sum_{y=1}^{N} m^{y \to x}\frac{I^y}{I^x}(f_A^x - f_A^y),$$

$$\frac{\mathrm{d}f_B^x}{\mathrm{d}t} = s_B^x f_B^x (1 - f_B^x) + s_A^x D^x + s_E^x f_B^x (1 - f_B^x)\frac{f_{AB}^x}{f_B^x} + (\mu_B^x + \rho_B^x)(1 - f_B^x) - (\mu_b^x + \rho_b^x)f_B^x - \sum_{y=1}^{N} m^{y \to x}\frac{I^y}{I^x}(f_B^x - f_B^y), \tag{19}$$

$$\frac{\mathrm{d}D^x}{\mathrm{d}t} = (s_A^x - s^x + s_B^x - s^x)D^x - (\mu^x + \rho^x)D^x + s_E^x f_{AB}^x f_{ab}^x - \sum_{y=1}^{N} m^{y \to x}\frac{I^y}{I^x}\left(D^x - D^y - (f_A^x - f_A^y)(f_B^x - f_B^y)\right).$$

System (19) contains a number of quantities that we now define in more detail. First, the (additive) selection coefficient for resistance to drugs $A$ and $B$ in population $x$ are defined as

$$s_A^x = r_{Ab}^x - r_{ab}^x \quad \text{and} \quad s_B^x = r_{aB}^x - r_{ab}^x, \tag{20}$$

respectively, while epistasis in population $x$ is $s_E^x = r_{AB}^x + r_{ab}^x - r_{Ab}^x - r_{aB}^x$. It follows that we can write each of the per-capita growth rates, $r_{ij}^x$, as

$$r_{ij}^x = r^x + \mathbf{1}_A(i)s_A^x + \mathbf{1}_B(j)s_B^x + \mathbf{1}_A(i)\mathbf{1}_B(j)s_E^x. \tag{21}$$

This is why $r_{ab}^x = r^x$ can be thought of as 'baseline' per-capita growth. We define the average selection for resistance in population $x$ as

$$s^x = s_A^x f_A^x + s_B^x f_B^x + s_E^x f_{AB}^x. \tag{22}$$

Note that the average per-capita growth rate in population $x$ is therefore $r^x + s^x$, that is, average per-capita growth rate is the sum of the 'baseline' per-capita growth rate and the average selection for resistance.

## Metapopulation LD and MDR

Next, consider metapopulation (or total) LD and MDR. First, let $p^x = I^x / \sum_{j=1}^{N} I^j$ be the fraction of total infections in population $x$. Then the metapopulation quantities equivalent to *Equations (16)* are

$$\bar{f}_i = \sum_{x=1}^{N} p^x f_i^x \quad \text{and} \quad \bar{f}_{ij} = \sum_{x=1}^{N} p^x f_{ij}^x. \tag{23}$$

The standard measure of linkage disequilibrium at the level of the metapopulation is

$$D_M = \bar{f}_{AB} - \bar{f}_A \bar{f}_B. \tag{24}$$

which in terms of the population level variables is

$$D_M \equiv \sum_{x=1}^{N} p^x D^x + \sum_{x=1}^{N} p^x f_A^x f_B^x - \left( \sum_{x=1}^{N} p^x f_A^x \right) \left( \sum_{x=1}^{N} p^x f_B^x \right) = \bar{D} + \text{cov}(f_A, f_B) \tag{25}$$

where $\bar{D}$ is the average population LD and $\text{cov}(f_A, f_B)$ is the spatial covariance between frequency of resistance to drug $A$ and frequency of resistance to drug $B$.

Using these variables, the three dynamical equations for studying metapopulation MDR are

$$
\begin{aligned}
\frac{d\bar{f}_A}{dt} &= \bar{s}_A \bar{f}_A (1 - \bar{f}_A) + \bar{s}_B D_M + \bar{s}_E \bar{f}_A (1 - \bar{f}_A) \frac{\bar{f}_{AB}}{\bar{f}_A} \\
&\quad + (\bar{\mu}_A + \bar{\rho}_A)(1 - \bar{f}_A) - (\bar{\mu}_a + \bar{\rho}_a)\bar{f}_A + \text{cov}(r, f_A) + \bar{f}_B \text{cov}\left( s_B, \frac{f_{AB}}{f_B} \right), \\
\frac{d\bar{f}_B}{dt} &= \bar{s}_B \bar{f}_B (1 - \bar{f}_B) + \bar{s}_A D_M + \bar{s}_E \bar{f}_B (1 - \bar{f}_B) \frac{\bar{f}_{AB}}{\bar{f}_B} \\
&\quad + (\bar{\mu}_B + \bar{\rho}_B)(1 - \bar{f}_B) - (\bar{\mu}_b + \bar{\rho}_b)\bar{f}_B + \text{cov}(r, f_B) + \bar{f}_A \text{cov}\left( s_A, \frac{f_{AB}}{f_A} \right), \\
\frac{dD_M}{dt} &= (\bar{s}_A - \bar{s} + \bar{s}_B - \bar{s})D_M - (\bar{\mu} + \bar{\rho})D_M + \bar{s}_E \bar{f}_{ab} \bar{f}_{AB} + \text{cov}(r, D) + \text{coskew}(r, f_A, f_B) \\
&\quad + \sum_{d \in \{A,B\}} (1 - \bar{f}_d) \bar{f}_d \text{cov}\left( s_d, \frac{f_{AB}}{f_d} \right) + (1 - \bar{f}_A)\Lambda_{Aa} - \bar{f}_A \Lambda_{aA} + (1 - \bar{f}_B)\Lambda_{Bb} - \bar{f}_B \Lambda_{bB}.
\end{aligned}
\tag{26}
$$

Note that in the equation $dD_M/dt$, there are terms involving $\Lambda_{ij}$ which we chose to neglect in *Equation (7)* given in the main text. These terms are

$$\Lambda_{Aa} = \text{cov}\left( \mu_A + \rho_A, \frac{f_{aB}}{1 - f_A} \right) \quad \text{and} \quad \Lambda_{aA} = \text{cov}\left( \mu_a + \rho_a, \frac{f_{AB}}{f_A} \right), \tag{27}$$

while

$$\Lambda_{Bb} = \text{cov}\left( \mu_B + \rho_B, \frac{f_{aB}}{1 - f_B} \right) \quad \text{and} \quad \Lambda_{bB} = \text{cov}\left( \mu_b + \rho_b, \frac{f_{AB}}{f_B} \right). \tag{28}$$

Thus the expression

$$(1 - \bar{f}_A)\Lambda_{Aa} - \bar{f}_A \Lambda_{aA} + (1 - \bar{f}_B)\Lambda_{Bb} - \bar{f}_B \Lambda_{bB} \tag{29}$$

in the equation $dD_M/dt$ is the effect upon $D_M$ of spatial heterogeneity in mutation and recombination rates ($\mu_i^x \neq \mu_i^y$ and/or $\rho_i^x \neq \rho_i^y$) coupled with differences in the proportion of infections with allele $i$ (e.g. $i = A$ or $i = a$) that are resistant to the other drug ($j = B$). In particular, populations in which infections are more likely to acquire resistance through mutation/recombination disproportionately affect metapopulation LD through an increase in doubly-resistant infections. However, these terms are likely to be quite small because they require that substantial differences in mutation/recombination rates exist between populations. Since these terms are unlikely to be a significant contributor to the dynamics of $D_M$, we ignore them in the main text.

There remains a number of other quantities in system (*Equation 26*) that we now define in more detail. First, the probability that an infection resistant to drug $d$ is found in population $x$ is

$$p^x \frac{f_d^x}{\bar{f}_d}. \tag{30}$$

For example, if we apply our variable definitions, it is straightforward to show that

$$p^x \frac{f_A^x}{\bar{f}_A} = \frac{I_{Ab}^x + I_{AB}^x}{\sum_{y=1}^N (I_{Ab}^y + I_{AB}^y)}. \tag{31}$$

Next, to compute the metapopulation-level selection coefficients, and mutation/recombination rates, we need to compute the weighted average of the population quantities, where the weights are the probability that an infection of a particular type is in population $x$ (calculated above). Applying this logic, the metapopulation-level selection coefficients and epistasis are

$$\bar{s}_i = \sum_{x=1}^N p^x \frac{f_i^x}{\bar{f}_i} s_i^x \quad \text{and} \quad \bar{s}_E = \sum_{x=1}^N p^x \frac{f_{AB}^x}{\bar{f}_{AB}} s_E^x. \tag{32}$$

The average selection for resistance in the metapopulation is

$$\bar{s} = \bar{s}_A \bar{f}_A + \bar{s}_B \bar{f}_B + \bar{s}_E \bar{f}_{AB}. \tag{33}$$

The per-capita mutation and recombination rates follow similarly. Recall that $\mu_\ell$ and $\rho_\ell$ are the per-capita rates at which infections gain allele $\ell$. Thus, for example,

$$\bar{\mu}_A = \sum_{x=1}^N p^x \frac{1 - f_A^x}{1 - \bar{f}_A} \mu_A^x \quad \text{and} \quad \bar{\mu}_a = \sum_{x=1}^N p^x \frac{f_A^x}{\bar{f}_A} \mu_a^x. \tag{34}$$

Similar calculations can be made to arrive at $\bar{\mu}_B$, $\bar{\mu}_b$, and the various $\bar{\rho}_\ell$. The total per-capita mutation and recombination rates are

$$\bar{\mu} = \bar{\mu}_a + \bar{\mu}_A + \bar{\mu}_b + \bar{\mu}_B \quad \text{and} \quad \bar{\rho} = \bar{\rho}_a + \bar{\rho}_A + \bar{\rho}_b + \bar{\rho}_B. \tag{35}$$

## Covariance and coskewness

Finally, we also use a number of covariance terms and a coskewness terms. Let $\mathbb{E}[c]$ denote the expectation of the quantity $c$. Then applying the definition of covariance, we have

$$\begin{aligned}
\text{cov}(f_A, f_B) &= \mathbb{E}[f_A f_B] - \mathbb{E}[f_A]\mathbb{E}[f_B] \\
&= \sum_{x=1}^N p^x f_A^x f_B^x - \left(\sum_{x=1}^N p^x f_A^x\right)\left(\sum_{x=1}^N p^x f_B^x\right)
\end{aligned}$$

Following the same procedure, we can calculate $\text{cov}(r, f_A)$ and $\text{cov}(r, D_M)$. When the covariance involves quantities that also specifically depend upon particular allele(s), the only difference is that when computing the expectation the probability used is the probability that an allele $\ell$ is in population $x$. For example,

$$\begin{aligned}
\text{cov}\left(s_A, \frac{f_{AB}}{f_A}\right) &= \mathbb{E}\left[s_A \frac{f_{AB}}{f_A}\right] - \mathbb{E}[s_A]\mathbb{E}\left[\frac{f_{AB}}{f_A}\right] \\
&= \sum_{x=1}^N p^x \frac{f_A^x}{\bar{f}_A} s_A^x \frac{f_{AB}^x}{f_A^x} - \left(\sum_{x=1}^N p^x \frac{f_A^x}{\bar{f}_A} s_A^x\right)\left(\sum_{x=1}^N p^x \frac{f_A^x}{\bar{f}_A} \frac{f_{AB}^x}{f_A^x}\right) \\
&= \sum_{x=1}^N p^x \frac{s_A^x f_{AB}^x}{\bar{f}_A} - \left(\sum_{x=1}^N p^x \frac{f_A^x}{\bar{f}_A} s_A^x\right)\left(\sum_{x=1}^N p^x \frac{f_{AB}^x}{\bar{f}_A}\right) \\
&= \sum_{x=1}^N p^x \frac{f_{AB}^x}{\bar{f}_A}(s_A^x - \bar{s}_A).
\end{aligned}$$

The covariance terms involving the recombination and mutation rates follow similarly, with the appropriate exchanges of variables. Finally, we have the coskewness term, which can be calculated as

$$\begin{aligned}
\text{coskew}(r, f_A, f_B) &= \mathbb{E}[(r - \mathbb{E}[r])(p_A - \mathbb{E}[f_A])(f_B - \mathbb{E}[f_B])] \\
&= \text{cov}(r, f_A f_B) - \bar{f}_B \text{cov}(r, f_A) - \bar{f}_A \text{cov}(r, f_B).
\end{aligned}$$

## Specific examples

### Equilibrium analysis of metapopulation consisting of independent populations

This is a version of one of the models presented in *Lehtinen et al., 2019*. The metapopulation consists of $N$ populations. The populations are independent (i.e, there is no migration between populations), and each population is assumed to be of a fixed size of unity, so $S^x = 1 - \sum_{ij} I_{ij}^x$. Resistance is gained and lost through unbiased mutation occurring at rate μ and there is no recombination. Therefore

$$\frac{\mathrm{d}I_{ij}^x}{\mathrm{d}t} = \left( \beta_{ij}^x S^x - \alpha_{ij}^x - \mathbf{1}_a(i)\tau_A^x - \mathbf{1}_b(j)\tau_B^x - (1 - \mathbf{1}_A(i)\mathbf{1}_B(j))\tau_{AB}^x \right) I_{ij}^x + \mu\left( \sum_\ell (I_{\ell j}^x + I_{i\ell}^x) - 4I_{ij}^x \right). \tag{36}$$

Let $\Delta z_d^x$ and $\Delta z_E^x$ denote the contribution of parameter $z$ to the additive selection coefficient (for drug $d$-resistance) and epistasis, respectively, in population $x$. Specifically,

$$\begin{aligned}
\Delta\beta_A^x = \beta_{Ab}^x - \beta_{ab}^x, &\quad \Delta\beta_B^x = \beta_{aB}^x - \beta_{ab}^x, &\quad \Delta\beta_E^x = \beta_{AB}^x + \beta_{ab}^x - \beta_{Ab}^x - \beta_{aB}^x \\
\Delta\alpha_A^x = \alpha_{Ab}^x - \alpha_{ab}^x, &\quad \Delta\alpha_B^x = \alpha_{aB}^x - \alpha_{ab}^x, &\quad \Delta\alpha_E^x = \alpha_{AB}^x + \alpha_{ab}^x - \alpha_{Ab}^x - \alpha_{aB}^x.
\end{aligned} \tag{37}$$

Then if we let $r_{ij}^x$ denote the per-capita growth term of an $ij$-infection in subpopulation $x$ (the first term in brackets in *Equation (36)*), we can partition this as

$$r_{ij}^x = r^x + \mathbf{1}_A(i)s_A^x + \mathbf{1}_B(j)s_B^x + \mathbf{1}_A(i)\mathbf{1}_B(j)s_E^x \tag{38}$$

where

$$\begin{aligned}
r^x &= \beta_{ab}^x S^x - \alpha_{ab}^x - \tau_A^x - \tau_B^x - \tau_{AB}^x \\
s_A^x &= \Delta\beta_A^x S^x - \Delta\alpha_A^x + \tau_A^x \\
s_B^x &= \Delta\beta_B^x S^x - \Delta\alpha_B^x + \tau_B^x \\
s_E^x &= \Delta\beta_E^x S^x - \Delta\alpha_E^x + \tau_{AB}^x
\end{aligned} \tag{39}$$

This notation and formulation differs from that of *Lehtinen et al., 2017*; *Lehtinen et al., 2019* in that they assumed costs were multiplicative, that is,

$$\beta_{ab}^x = \beta^x, \quad \beta_{Ab}^x = \beta^x c_{\beta_A}^x, \quad \beta_{aB}^x = \beta^x c_{\beta_B}^x, \quad \beta_{AB}^x = \beta^x c_{\beta_A}^x c_{\beta_B}^x \tag{40}$$

and

$$\alpha_{ab}^x = \alpha^x, \quad \alpha_{Ab}^x = \frac{\alpha^x}{c_{\alpha_A}^x}, \quad \alpha_{aB}^x = \frac{\alpha^x}{c_{\alpha_B}^x}, \quad \alpha_{AB}^x = \frac{\alpha^x}{c_{\alpha_A}^x c_{\alpha_B}^x} \tag{41}$$

where $0 \le c_{\beta_\ell}^x \le 1$ and $0 \le c_{\alpha_\ell}^x \le 1$ (note the slightly different notation used for multiplicative costs in *1*). There are two consequences of multiplicative costs . First, multiplicative costs produce epistasis. For the model of *Lehtinen et al., 2017*; *Lehtinen et al., 2019*:

$$s_E^x = \beta^x(1 - c_{\beta_A}^x)(1 - c_{\beta_B}^x)S^x - \alpha^x \frac{(1 - c_{\alpha_A}^x)(1 - c_{\alpha_B}^x)}{c_{\alpha_A}^x c_{\alpha_B}^x} + \tau_{AB}^x. \tag{42}$$

Thus, in this model, there exists epistasis whenever there is a cost of resistance or drugs are prescribed in combination, $\tau_{AB}$. More specifically, transmission costs and combination treatment will produce positive epistasis, while duration of carriage costs will produce negative epistasis. Inclusion of epistasis (through multiplicative costs) is not necessarily a problem, and for epidemiological reasons multiplicative costs may be preferable. Indeed, because epistasis plays a central role in multilocus dynamics, it is valuable to recognize if/when epistasis is occurring. However, our analysis in the main text focused upon how population variation in susceptible densities can create correlations in the selection coefficients, favoring MDR, and so we excluded the possibility of epistasis.

The second consequence of multiplicative costs is that they have implications for the cost of resistance within a population. For the above model, and assuming $c_{\alpha_A}^x = c_{\alpha_A}$, $c_{\beta_A}^x = c_{\beta_A}$, the selection coefficient for resistance to drug $A$ can be written

$$s_A^x = \tau_A^x - \alpha^x \frac{1 - c_{\alpha_A}}{c_{\alpha_A}} - \beta^x \frac{1 - c_{\beta_A}}{c_{\beta_A}} S^x. \tag{43}$$

From *Equation (43),* we see that the costs of resistance depend upon the population's epidemiological parameters (by assumption). Specifically, ignoring concomitant effects upon $S^x$, populations that are more transmissible (larger $\beta^x$) with a shorter duration of carriage (larger $\alpha^x$) pay higher costs of resistance due to how the cost parameters interact with the epidemiological parameters. For example, if there were no costs to transmission ($c_{\beta_A} = 1$), then *Equation (43)* predicts that populations with longer duration of carriage (smaller $\alpha^x$) are more likely to become resistant, because they pay disproportionately lower costs. Indeed, if the populations represent serotype, than this is an example of epistasis between serotype and resistance, which favors LD between duration of carriage and resistance.

To put this in a biological context, if (for example) the populations correspond to the different capsular serotypes of *S. pneumoniae*, it is possible that the differences between capsules interact with the mechanism of resistance so as to make resistance more costly for more transmissible capsular serotypes or those capsular serotypes associated with longer duration of carriage (multiplicative costs), but it is also possible that no interaction occurs between the capsule differences and the mechanism of resistance (additive costs) or that resistance is less costly for more transmissible serotypes or for capsular serotypes associated with a shorter duration of carriage.

Irrespective of whether the costs are multiplicative or additive, we would attach the constraints that $0 \leq \beta_{ij}^x \leq \beta_{ab}^x$ and $0 \leq \alpha_{ab}^x \leq \alpha_{ij}^x$, that is, carriage of one or more resistance alleles will never increase transmissibility or decrease duration of carriage, respectively; otherwise there are no costs. As an example, if we were interested in attaching additive resistance costs to transmission, if we let $c_{\beta_d}^x$ and $c_{\beta_{AB}}^x$ be the (additive) transmission cost of resistance to drug $d$ and epistatic transmission cost, respectively, in population $x$, so that

$$\beta_{ij}^x = \beta_{ab}^x - \mathbf{1}_A(i) c_{\beta_A}^x - \mathbf{1}_B(i) c_{\beta_B}^x - \mathbf{1}_{AB}(ij) c_{\beta_{AB}}^x, \tag{44}$$

then we could attach the constraints

$$\beta_{ij}^x = \min(0, \beta_{ab}^x - \mathbf{1}_A(i) c_{\beta_A}^x - \mathbf{1}_B(i) c_{\beta_B}^x - \mathbf{1}_{AB}(ij) c_{\beta_{AB}}^x), \tag{45}$$

or simply assume that

$$c_{\beta_A}^x + c_{\beta_B}^x + |c_{\beta_{AB}}^x| \leq \beta_{ab}^x. \tag{46}$$

There are many possible ways of implementing the constraints. One point to note is that even with additive costs, the choice of constraints can also potentially create epistasis; this could occur in the case of *Equation (45)*.

In *Figure 3*, we consider three scenarios; whenever possible we choose parameter values to agree with those of Figure 4 in *Lehtinen et al., 2019*. In each scenario, we assume there are 20 independent populations, that the per-capita mutation rate is $\mu = 10^{-10}$, and there is no epistasis, $s_E^x = 0$. In subplot 3a, we set $\beta_{ab}^x = 2$, while duration of carriage varies among populations from $\alpha_{ab}^x = 0.25$ to $\alpha_{ab}^x = 1.75$. In subplot 3b we set $\alpha_{ab}^x = 0.5$, while transmission varies among populations from $\beta_{ab}^x = 1$ to $\beta_{ab}^x = 3$. In both subplots 3a and 3b, $\Delta\alpha_A^x = \Delta\alpha_B^x = 0$, while $\Delta\beta_A^x = \Delta\beta_B^x = -0.1$. Finally in subplot 3 c, $\Delta\beta_A^x = \Delta\beta_B^x = 0$, while duration of carriage varies among populations from $\alpha_{ab}^x = 0.25$ to $\alpha_{ab}^x = 1.75$, with $\Delta\alpha_A^x = \Delta\alpha_B^x = 0.05$.

## Transient dynamics and MDR in *Streptococcus pneumoniae*

Here we use a variant of the model originally proposed by *Lehtinen et al., 2017*, *Lehtinen et al., 2019* in which the populations represent different serotypes. Resistance is gained and lost through unbiased mutation at a per-capita rate $\mu$ and there is no recombination of resistance loci.

Applying these assumptions and using the notation presented with our model from the main text, this yields

$$\frac{\mathrm{d}I_{ij}^x}{\mathrm{d}t} = \left( \beta_{ij}^x \nu(I,x)S - \alpha_{ij}^x - \mathbf{1}_a(i)\tau_A - \mathbf{1}_b(j)\tau_B - (1 - \mathbf{1}_A(i)\mathbf{1}_B(j))\tau_{AB} \right)I_{ij}^x + \mu\left( \sum_\ell (I_{\ell j}^x + I_{i\ell}^x) - 4I_{ij}^x \right) \quad (47)$$

where

$$\nu(I,x) = \left( 1 - \left[ \frac{\sum_{ij} I_{ij}^x}{\sum_{k=1}^N \sum_{ij} I_{ij}^k} - \frac{1}{N} \right] \right)^\omega \quad (48)$$

is a balancing function intended to mimic the stabilizing effect adaptive host immunity has upon serotype diversity ($\omega$ controls the strength of this effect; see *Lehtinen et al., 2017*; *Lehtinen et al., 2019*). The treatment rates in *Equation (47)* are assumed to be independent of serotype. Note that although we could mechanistically model the susceptible hosts available to each subpopulation, $S^x$, it is more straightforward and computationally simpler to use the phenomenological model given above in which $S^x = \nu(I,x)S$. The function $\nu(I,x)$ ensures that $S^x$ has the two properties we are interested in: (i) there can be variation across populations of available susceptibles, and (ii) this variation is linked to population attributes (e.g. transmissibility and duration of carriage). The primary conclusions of our analysis would hold if a mechanistic model for $S^x$ were used instead.

If we let $r_{ij}^x$ denote the per-capita growth term of an $ij$-infection belonging to serotype $x$ (the first term in brackets in *Equation (47)*), we can partition this as

$$r_{ij}^x = r^x + \mathbf{1}_A(i)s_A^x + \mathbf{1}_B(j)s_B^x + \mathbf{1}_A(i)\mathbf{1}_B(j)s_E^x \quad (49)$$

where if we use the notation introduced in *Equation (37)*,

$$\begin{aligned} r^x &= \beta_{ab}^x \nu(I,x)S - \alpha_{ab}^x - \tau_A - \tau_B - \tau_{AB} \\ s_A^x &= \Delta\beta_A^x \nu(I,x)S - \Delta\alpha_A^x + \tau_A \\ s_B^x &= \Delta\beta_B^x \nu(I,x)S - \Delta\alpha_B^x + \tau_B \\ s_E^x &= \Delta\beta_E^x \nu(I,x)S - \Delta\alpha_E^x + \tau_{AB} \end{aligned} \quad (50)$$

For simplicity, we keep total population size constant, and so set $S = 1 - \sum_{x=1}^N \sum_{ij} I_{ij}^x$.

The simulations in *Figure 5* assume the metapopulation is initially treated at per-capita rates $(\tau_A, \tau_B, \tau_{AB}) = (0.12, 0, 0)$, until $t = 1000$ when these rates switch to $(\tau_A, \tau_B, \tau_{AB}) = (0.07, 0.1, 0)$. Other parameters values used are $N = 12$, $\omega = 3$, $\Delta\beta_A^x = \Delta\beta_B^x = -0.2$, $\Delta\alpha_A^x = \Delta\alpha_B^x = 0.05$ and $\mu = 10^{-8}$. Finally, because *Streptococcus* serotypes differ based upon duration of carriage and transmissibility, and there is evidence of a positive correlation between the two (*Weinberger et al., 2009*; *Zafar et al., 2017*), $\alpha_{ab}^x$ was chosen to assume evenly spaced parameter values from $\alpha_{ab}^x = 0.2$ to $\alpha_{ab}^x = 0.7$, while $\beta_{ab}^x$ was chosen to assume evenly spaced parameter values from $\beta_{ab}^x = 3.25$ to $\beta_{ab}^x = 3$. Cost epistasis, when it is present, is assumed to solely effect transmissibility (i.e. $\Delta\alpha_E^x = 0$). When there is positive epistasis, $\Delta\beta_E^x = 0.065$, whereas for negative epistasis, $\Delta\beta_E^x = -0.015$.

## Contrasting drug prescription strategies in a hospital-community setting

When we model the hospital and community, we use *Equation (2)* and assume the susceptible host density is controlled by

$$\begin{aligned} \frac{\mathrm{d}S^x}{\mathrm{d}t} = {}& \theta^x - dS^x - m^{x\to y}S^x + m^{y\to x}S^y - \sum_{ij}\beta_{ij}^x I_{ij}^x S^x \\ & + \sum_{ij}(\alpha_{ij}^x - d)I_{ij}^x + \sum_{ij}\left(\mathbf{1}_a(i)\tau_A^x + \mathbf{1}_b(j)\tau_B^x + (1 - \mathbf{1}_A(i)\mathbf{1}_B(j))\tau_{AB}^x\right)I_{ij}^x \end{aligned} \quad (51)$$

where $\theta^x$ is the influx of new hosts and $d$ is the background mortality rate.

In the hospital/community model, we assume population $C$ is the 'community' and population $H$ is the 'hospital'. Therefore, we let $\theta^H = 0$, and $m^{C\to H} = m/\sum_{ij} I_{ij}^C$ be the rate at which individuals are admitted to the hospital, which is independent of population size. Individuals exit the hospital at a

constant rate $m^{H \rightarrow C}$, so they spend on average $1/m^{H \rightarrow C}$ time units in hospital (assuming background mortality is low). The specification of the migration rates in this way allows us to ensure the 'community' is always much larger than the 'hospital'. In *Figure 6*, we assumed that the total prescription rate per population, $\tau^C$ or $\tau^H$, was the same for each strategy, and that drugs are prescribed at $15\times$ the rate in the hospital versus the community, that is, $\tau^H = 15\tau^C$. For 'mixing', this means $(\tau_A^x, \tau_B^x, \tau_{AB}^x) = (\tau^x/2, \tau^x/2, 0)$, whereas for 'combination', this means $(\tau_A^x, \tau_B^x, \tau_{AB}^x) = (0, 0, \tau^x)$. Finally, for 'cycling' drug $A$ and $B$ were rotated from hospital to community every 50 time units so that either $(\tau_A^x, \tau_B^x, \tau_{AB}^x) = (\tau^x, 0, 0)$ or $(\tau_A^x, \tau_B^x, \tau_{AB}^x) = (0, \tau^x, 0)$ depending on the rotation. Therefore in *Figure 6* the period is of length $T = 100$. In *Figure 6*, we numerically integrate the system until $t = 10^4$; the final state at $t = 10^4$ is then what is plotted for the combination and mixing scenarios, whereas for cycling we plot the average state across the last two rotations (i.e. the average state over the final period, from $t = 9,900$ to $t = 10^4$). *Figure 7* shows how changing the length of time between drug rotations affects the evolution of single- and multi-drug resistance. Specifically, when drug rotations are frequent (with period of $T = 1$), cycling behaves like mixing and so positive LD is produced (*Figure 7a*). As drug rotations become less frequent (period of $T = 24$ and $T = 160$), cycling generates a negative covariance in selection, which in turn produces negative LD (*Figure 7a*). Thus, when drug rotations are more frequent, single-drug resistance is delayed and emerges at higher treatment rates, but the evolution of MDR occurs at lower treatment rates (*Figure 7b*). When drug rotations are infrequent, single-drug resistance emerges at lower treatment rates, but MDR evolution is delayed, emerging at higher treatment rates (*Figure 7b*). Parameters used in *Figure 6* and *Figure 7* were $\beta_{ab}^x = 2$, $\Delta\beta_A^x = \Delta\beta_B^x = -0.4$, $\alpha_{ab}^x = 0.1$, $\Delta\alpha_A^x = \Delta\alpha_B^x = 0.02$, $d = 0.01$, $\theta^C = 0.2$, $\theta^H = 0$, $m^{H \rightarrow C} = 0.5$, $m = 0.2$, $\mu = 10^{-7}$, $\sigma = 0$.

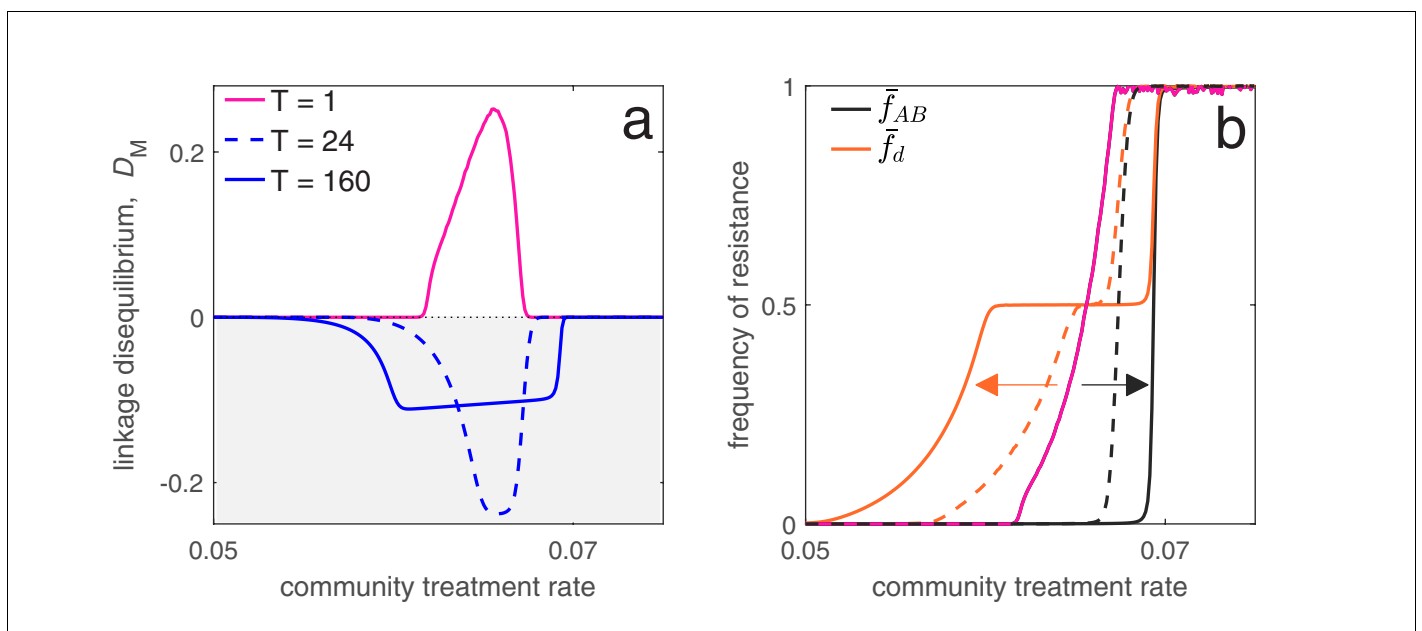

**Figure 7.** Time between drug rotations affects the evolution of both single- and multi-drug resistance. When drugs are rotated every 0.5 time units (period of $T = 1$; magenta curves), cycling behaves like mixing and positive LD is generated. As we increase the time between rotations (period of $T = 24$ and $T = 160$), a negative covariance in selection is generated, producing negative LD (dashed and solid blue curves). In panel a we show the metapopulation LD, $D_\mathrm{M}$, while in panel b, we show the frequency of resistance in the metapopulation. When drugs are rotated frequently, single drug resistance emerges at higher treatment rates but MDR emerges at lower treatment rates, as compared to when drugs are rotated infrequently. Thus, there is a trade-off (indicated by the arrows in panel b) associated with time between drug rotations: we can delay single drug resistance but promote MDR (frequent drug rotations), or delay MDR but promote single drug resistance (infrequent drug rotations). In all cases, we integrated the system until $t = 10^4$, then averaged the system state over the final two rotations (i.e. over a single period). The remaining parameter values are provided in Materials and methods 'Contrasting drug prescription strategies in a hospital-community setting'.

## Acknowledgements

We thank Sonja Lehtinen, Brian J Arnold and Stephen M Kissler for helpful comments. This work was supported by a NSERC-CRSNG postdoctoral fellowship to DVM. SG acknowledges support from the Agence Nationale de la Recherche, grant ANR-17-CE35-0012 'EVOMALWILD'.

## Additional information

### Funding

| Funder | Grant reference number | Author |
| --- | --- | --- |
| Natural Sciences and Engineering Research Council of Canada | Postdoctoral fellowship | David V McLeod |
| Agence Nationale de la Recherche | ANR-17-CE35-0012 | Sylvain Gandon |

The funders had no role in study design, data collection and interpretation, or the decision to submit the work for publication.

### Author contributions

David V McLeod, Conceptualization, Formal analysis, Investigation, Visualization, Methodology, Writing - original draft, Writing - review and editing; Sylvain Gandon, Conceptualization, Supervision, Investigation, Methodology, Writing - review and editing

### Author ORCIDs

David V McLeod https://orcid.org/0000-0003-2551-7877

### Decision letter and Author response

Decision letter https://doi.org/10.7554/eLife.65645.sa1
Author response https://doi.org/10.7554/eLife.65645.sa2

## Additional files

### Supplementary files

• Transparent reporting form

### Data availability

All data used was from a previously published study (Lehtinen et al. 2019 PLoS Pathogens and Turner et al. 2012 PLoS ONE); this data has been uploaded by those authors to a public repository, we downloaded it from there and have provided the details.

The following previously published dataset was used:

| Author(s) | Year | Dataset title | Dataset URL | Database and Identifier |
| --- | --- | --- | --- | --- |
| Lehtinen S, Blanquart F, Croucher NJ, Turner P, Lipsitch M, Frasera C | 2019 | S1 File. Resistance profiles and duration of carriage estimates for the Maela dataset. | https://doi.org/10.1371/journal.ppat.1007763.s002 | PLoS Pathogens, 10.1371/journal.ppat.1007763.s002 |

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
