## [Decision Letter]

**Acceptance summary:**

This paper addresses the important question of multidrug resistance evolution, which is of both theoretical and applied interest. The authors' efforts to carefully distinguish population and metapopulation linkage disequilibrium and to develop a framework to rigorously analyze the relationship between the two represent an advance in our understanding of microbial population dynamics.

**Decision letter after peer review:**

[Editors’ note: the authors submitted for reconsideration following the decision after peer review. What follows is the decision letter after the first round of review.]

Thank you for submitting your work entitled "Understanding the evolution of multiple drug resistance in structured populations" for consideration by *eLife*. Your article has been reviewed by 2 peer reviewers, one of whom is a member of our Board of Reviewing Editors, and the evaluation has been overseen by a Senior Editor. The following individual involved in review of your submission has agreed to reveal their identity: Brian J Arnold (Reviewer #2).

Our decision has been reached after consultation between the reviewers. Based on these discussions and the individual reviews below, we regret to inform you that your work will not be considered further for publication in *eLife*.

We have appended to this message a summary of the two reviewers' comments, including both major and minor concerns. Briefly, there were concerns about the extent of the claims made about the model explaining features of data, the clarity of the model development and notation, and links between the results and previous literature. Please see below for details.

*Reviewer 1:*

MDR is more common than expected by random chance, especially in particular bacterial species. Understanding how this linkage between resistance elements arises is important for public health but also for generally making sense of genomic data, e.g. classifying genomic patterns as outcomes of specific biological processes. Here, McLeod and Gandon show that spatial heterogeneity across subdivided populations can create LD between resistance elements, even in the absence of epistasis.

The novelty of this manuscript appears to be an extension of Day and Gandon, (2012) to a metapopulation model as well as comparing output of the model to a dataset on pneumococcus, which has sort of served as a model for the ecology and evolution of MDR. One particular result of interest was the observation of simulated transient LD even without epistasis. Though this particular observation is not necessarily new (Martin et al., 2006; Day and Gandon, 2012), casting this metapopulation model in an epidemiological framework appears to be new to my knowledge.

However, the intuition this model provides for D_tot_ > 0 at equilibrium is intriguing: variability in susceptibility density across populations can create covariance in selection coefficients for the two alleles, in turn creating correlations between subpopulation allele frequencies, driving D_tot_ > 0. This insight is extremely general and has nothing to do with the biology of a particular species.

While this isn't a major concern, I wanted to point out what I thought was confusing language that made it take longer for me to understand the results.

1) I found the number of ways to quantify LD a little confusing and done with ambiguous language. D is first used in it's standard form D = P_ab_ – P_a_*P_b_ (Equation 1), which made it confusing when I then encountered D within the definition of D_tot_ (Equation 5), where D is the weighted average of D = P_ab_ – P_a_*P_b_ for each of the subpopulations. I think it would be clearer if D perhaps consistently referred to a subpopulation, the average D across subpopulations were instead indicated with a summation in Equation 5.

Also with Equation 5, it took me a while to figure out what cov(p_a_, p_b_) actually referred to, as the definition "spatial covariance between resistance to drugs A and B", along with being primed by historical literature, made me think of this covariance as LD (i.e. how frequently alleles are found within the same genome). However, this is referring to covariance of *allele frequencies across subpopulations* and has nothing to do with whether or not they're found in the same genome. More specific/clear language will help readers distinguish between the different covariances studied here: between alleles across genomes (LD) and between allele frequencies across subpopulations (cov(p_a_, p_b_)).

2) Also, I'm left not knowing the ultimate conclusion of applying the model to the pneumococcal data. On line 226 you claim that transient dynamics and epistasis can explain patterns in pneumococcus, whereas on line 301 you claim that variation in S^X^ across subpopulations can generate LD. Some short conclusion regarding these two claims would be useful.

Citations:

Day & Gandon, (2012). The evolutionary epidemiology of multilocus drug resistance. Evolution. https://doi.org/10.1111/j.1558-5646.2011.01533.x

Martin, Otto & Lenormand, (2006). Selection for recombination in structured populations. Genetics, 172(1), 593-609. https://doi.org/10.1534/genetics.104.039982

1) For the case of additive selection, you provide one particular example of parameters that give rise to transient positive LD (Figure 4), but it would seem that many situations would also give rise to negative LD. Is there a symmetry to the model such that if you were to "integrate" across all possible starting conditions, the expectation would be zero? Or is there an asymmetry that makes transient LD more likely to be positive? I assume the latter if you add epistasis, but it's unclear with additive selection. This would help map intuition from your model to results such as Figure 3, in which it seems there's a skew towards positive LD. Is epistasis absolutely needed to explain this skew?

2) One interesting implication of this work is what to expect when bacterial "populations" are either not defined at all or even incorrectly defined. Since Dtot is very positive under many scenarios (even with negative epistasis! Figure 4), you will observe an excess of positive LD between resistance elements (given these starting conditions in Figure 4, that is).

*Reviewer 2:*

This is an interesting and important topic and the paper promised to be a new take on it, from the point of view of dynamics of linkage disequilibrium.

My major concerns are (1) clarity of the mathematics, notation and model development and (2) with the claims that the model explains data, without a wide range of simulations or a direct comparison of model to data (for example see below re line 239 onwards).

The model in Equation (2) looks linear and the reader has to see figure 1 to note that there is actually a nonlinear term here (the susceptible * infectious term, and a product of two infectious terms) because the s_A_ etc depend on the prevalence.

The model derivation is cumbersome and far from straightforward; I was unable to directly derive (3) from (2). Instead, for the first three terms of (3) I obtained

dpA/dt = (1-pA)(pA sA + pAB (sAB – sA – sB) ) + sB D

instead of the expression in the paper which is

(1-pA) (pA sA + sAB pAB) + sB D

This would seem to carry through to the other equations. However my derivation attempts were simply using sA, sAB, sB as given in the text, and putting in the rate of growth of the AB variant (r + sAB) as in (2) for example, without substituting in the expressions from Figure 1 for sA, sB, and sAB. This may work it out (?) However this left me a bit sceptical.

Bottom of page 4: if doubly-resistant infections are overrepresented in the population, D^X^ > 0. < -- middle page 5 "since D^X^ > 0" Which is it?

Page 10: "…, drug B resistance is often more likely to occur in an infection with a genetic background resistant to drug A." -- more likely than what?

Line 239++ : This section makes the claim:

"Thus transient dynamics coupled with epistasis can explain the significant within-serotype LD observed in Streptococcus pneumoniae".

This is a big claim, based on a qualitative paragraph with no direct comparison to data, in which there is an underlying assumption of fixation of resistance to drug A in many serotypes. To my knowledge we often do not see fixation of resistance in Streptococcus pneumoniae or in the Maela data in particular, but rather long-term coexistence. I am not convinced that the dynamics of this model explain the significant LD in the data as claimed.

Equation (2) – there is a note above that rho_l_^X^ depends on infection densities; the nature of this dependence should be specified.

Page 12: why, if there is no mechanism maintaining within-population diversity, does this necessarily mean D^X^ = 0 (line 282) ?

[Editors’ note: further revisions were suggested prior to acceptance, as described below.]

Thank you for submitting your article "Understanding the evolution of multiple drug resistance in structured populations" for consideration by *eLife*. Your article has been reviewed by 2 peer reviewers, and the evaluation has been overseen by a Reviewing Editor and George Perry as the Senior Editor. The following individuals involved in review of your submission have agreed to reveal their identity: Sonja Lehtinen (Reviewer #1); Stephen M Kissler (Reviewer #2).

Essential revisions:

While the reviewers of the previous version of your paper were not available to reconsider the revised manuscript, the reviewers of this version of the paper did take the previous comments and your responses into consideration when composing their own reviews.

Collectively we see much potential in your manuscript, but we request the following points to be addressed at minimum (see below reviews for further details and more suggestions):

1. Expanded explanation of the assumptions behind the LD framework. In particular:

a) Is epistasis necessarily defined in terms of an additive expectation on growth rate? (If this is a standard result in population genetics, a reference to an accessible explanation would be enough).

b) The s coefficients are dependent on variable densities – does this matter for the partitioning and interpretation of equations 3 and 4?

2. If the authors disagree with Reviewer 1's questions about the interpretation that variation in susceptible density explains the effect in example 1 and in Lehtinen et al., 2019, more explanation is needed for the following points:

a) Why variation in clearance rate affects selection for resistance when there is a multiplicative cost on clearance rate, but no cost on transmission rate (for the re-interpretation of Lehtinen specifically)?

b) What is happening in example 2, where the effect is currently explained in terms of serotype-specific susceptible density, but the model does not include serotype-specific susceptibles?

3. For examples 2 and 3, please either explain why additive transmission costs are reasonable (given the concerns highlighted below) or change the modeling of cost.

*Reviewer #1 (Recommendations for the authors):*

My major scientific recommendations are covered by my review. There are some additional less fundamental points that I am happy to share with the authors if they are interested but won't include here as this review is already quite long.

In terms of presentation, I thought everything was mostly very clear, with the exception of the section on serotype dynamics. Here it would be more helpful to make clear the shift from a structured susceptible population to a shared susceptible population.

*Reviewer #2 (Recommendations for the authors):*

I was asked to review this manuscript after it had already gone through a round of revisions. My comments reflect both my own initial reading of the manuscript and my assessment of the authors' responses to the previous reviews.

While I did not have access the authors' initial submission, it appears that the authors have sufficiently addressed the previous reviewers' comments.

[Editors’ note: further revisions were suggested prior to acceptance, as described below.]

Thank you for resubmitting your work entitled "Understanding the evolution of multiple drug resistance in structured populations" for further consideration by *eLife*. Your revised article has been evaluated by George Perry (Senior Editor) and a Reviewing Editor.

The manuscript has been improved but there are some remaining issues that need to be addressed as outlined below in edited comments from Reviewer Sonja Lehtinen, as outlined in points 2-5 below. In addition, I have reviewed the discussion with the *eLife* editorial office about the article structure and have considered this question myself, ultimately coming to point 1.

1. Please do create a methods section for this paper. There is no need to remove anything from your current main text. Rather, in the new methods section please first overview the methods even if they are already described in more detail elsewhere in the paper (some repetition is ok, and I agree with the level of detail you have provided in the Results section, given the type of paper) in one or more subheadings, and then import all of the sub-headings and the full content of the appendix. Since we don't have page limitations and since this is important to the paper, I think it should be included in the methods section of the main text of the paper, rather than as an appendix.

2. The point the authors are making about density of susceptibles and that this point is not limited to additive costs is understood. However, the section on equilibrium patterns of MDR still should be updated because the difference between additive and multiplicative costs is not explicitly flagged in the main text. (I think the discussion in the Sup Mat is very helpful, but most readers will not read the Supplement). Specifically, when considering equilibrium patterns of MDR, the paragraph starting line 283 is only true for additive costs. Without reading the supplement, this paragraph comes across as a general result. If costs are multiplicative, variation in duration of carriage is both necessary (i.e. variation in transmission rate does not produce LD at equilibrium) and sufficient (explicit transmission costs are not needed). Thus, overall, it is true to say that variation in duration of carriage is not necessary, but not that it is not sufficient. It would be helpful to explicitly make this difference clear somewhere in this section. Please also rephrase the sentence lines 270-273 to avoid implying that the costs in Lehtinen et al., are additive.

3. The main text would benefit from a discussion of additive costs. Specifically:

a. The authors make a good point that cost parameters are subject to constraints however they are modelled. The reviewer's point was that these constraints are qualitatively different for additive transmission costs: reasonable additive costs for single resistance can lead to negative transmission rate for dual resistance (as in the example in my review) in the absence of cost epistasis. This suggests that for cost parameters where this is the case, assuming no epistasis cannot be appropriate. Although the specific parameter values the authors use don't give rise to this problem, the authors also present more general results for a model with additive costs and no epistasis, so it would be helpful to highlight this constraint somewhere.

b. The authors' careful consideration of how the specification of costs affects interpretation is indeed very useful and interesting. In addition to this conceptual point, the authors say that they are aiming to highlight mechanisms – e.g. variation in transmission rate – that could plausibly give rise to patterns of MDR. In the case of equilibrium patterns, variation in transmission rate only gives rise to LD when transmission costs are additive. The plausibility of variation in transmission rate as an explanation for equilibrium patterns of MDR therefore depends on the plausibility of additive transmission costs. Therefore, if the authors want to suggest that variation in transmission rate is a plausible explanation for patterns of MDR, it is necessary to include a discussion of whether/how additive transmission costs might arise.

4. Add "per carriage episode" to the summary of the biological interpretation of the duration of carriage effect in Lehtinen et al., 2017/2019 (lines 244-245), to make it clear that they were not suggesting that (host) populations with longer durations of pathogen carriage have greater antibiotic exposure. (This point was missed in the original review).

5. One final observation, which is not an essential revision and the authors should only address it if they think it will improve the paper. Points 2 and 3 are both related to the result that a model with a multiplicative transmission cost predicts, at equilibrium, LD between duration of carriage and resistance, but not transmission rate and resistance. Would the authors explain this in terms of variation in the density of susceptibles giving rise to the LD with duration of carriage, but this effect being offset by the epistatic interaction between transmission rate and cost of resistance leading to no LD between transmission rate and resistance? This might be interesting to explicitly discuss in the manuscript.

---

## [Author Response]

[Editors’ note: the authors resubmitted a revised version of the paper for consideration. What follows is the authors’ response to the first round of review.]

Essential revisions:Reviewer 1:MDR is more common than expected by random chance, especially in particular bacterial species. Understanding how this linkage between resistance elements arises is important for public health but also for generally making sense of genomic data, e.g. classifying genomic patterns as outcomes of specific biological processes. Here, McLeod and Gandon show that spatial heterogeneity across subdivided populations can create LD between resistance elements, even in the absence of epistasis.The novelty of this manuscript appears to be an extension of Day and Gandon, (2012) to a metapopulation model as well as comparing output of the model to a dataset on pneumococcus, which has sort of served as a model for the ecology and evolution of MDR. One particular result of interest was the observation of simulated transient LD even without epistasis. Though this particular observation is not necessarily new (Martin et al., 2006; Day and Gandon, 2012), casting this metapopulation model in an epidemiological framework appears to be new to my knowledge.However, the intuition this model provides for D^tot^ > 0 at equilibrium is intriguing: variability in susceptibility density across populations can create covariance in selection coefficients for the two alleles, in turn creating correlations between subpopulation allele frequencies, driving D_tot_ > 0. This insight is extremely general and has nothing to do with the biology of a particular species.

Yes, one of the main insights we gain from our analysis is a clarification and a generalisation of the results discussed recently by Lehtinen et al., (2019) about the conditions favouring MDR at equilibrium. We believe our Equation (5) provides a new and useful way to understand the evolution of MDR. We would also like to highlight that the dynamical equations we derive are very general and can be used to understand, and perhaps limit, the spread of MDR. For instance, the final example regarding the impact of cycling antibiotics between a hospital and community is new (previous studies did not study the influence of cycling drugs in a metapopulation on MDR evolution) and opens up new perspectives on the management of MDR resistance.

In our revisions, we extended and rewrote our discussions of the examples and we show more explicitly how equations 3 to 7 can help understand the complex evolutionary dynamics of drug resistance.

While this isn't a major concern, I wanted to point out what I thought was confusing language that made it take longer for me to understand the results.1) I found the number of ways to quantify LD a little confusing and done with ambiguous language. D is first used in its standard form D = P_ab_ – P_a_*P_b_ (Equation 1), which made it confusing when I then encountered D within the definition of D_tot_ (Equation 5), where D is the weighted average of D = P_ab_ – P_a_*P_b_ for each of the subpopulations. I think it would be clearer if D perhaps consistently referred to a subpopulation, the average D across subpopulations were instead indicated with a summation in Equation 5.Also with Equation 5, it took me a while to figure out what cov(p_a_, p_b_) actually referred to, as the definition "spatial covariance between resistance to drugs A and B", along with being primed by historical literature, made me think of this covariance as LD (i.e. how frequently alleles are found within the same genome). However, this is referring to covariance of *allele frequencies across subpopulations* and has nothing to do with whether or not they're found in the same genome. More specific/clear language will help readers distinguish between the different covariances studied here: between alleles across genomes (LD) and between allele frequencies across subpopulations (cov(p_a_, p_b_)).

We agree and we have rewritten many sections of the manuscript with this comment in mind. Additionally, we changed the notation to try improve the readability. We believe this rewording has significantly increased the clarity of our arguments.

2) Also, I'm left not knowing the ultimate conclusion of applying the model to the pneumococcal data. On line 226 you claim that transient dynamics and epistasis can explain patterns in pneumococcus, whereas on line 301 you claim that variation in S^X^ across subpopulations can generate LD. Some short conclusion regarding these two claims would be useful.

We have rewritten both of these examples in light of the reviewers’ comments, and hope the conclusions now are clearer. Specifically, variation in S^x^ (density of susceptibles) across populations can generate metapopulation LD, and in fact can lead to MDR overrepresentation in the metapopulation. However, this example is focused upon an equilibrium analysis. Consequently, although metapopulation LD exists at equilibrium, because nothing maintains diversity within-populations (i.e., within-serotypes), there will be no LD within serotypes at equilibrium. This would be true with or without epistasis; i.e., with epistasis, we would expect at equilibrium that $D^x^ = 0$ even if $D_M_ \not = 0$.

In the next example (the transient + epistasis example), when we consider transient dynamics we are specifically focused upon mechanisms that can generate (transient) LD within-serotype, i.e., $D^x^ \not=0$. As we detail in this example, there are two possibilities: epistasis or migration (serotype recombination or serotype switching), however migration (serotype recombination) is an unlikely explanation because the rate at which this occurs would have to be unrealistically high. Therefore, we focus upon exploring the potential role played by epistasis, and show that it can (transiently) generate LD within-serotype. Of course, since there is nothing maintaining within-host diversity within-serotype, at equilibrium $D^x^ = 0$ as in the previous example. We believe that the way these examples have been rewritten makes it clearer what we are trying to show.

1) For the case of additive selection, you provide one particular example of parameters that give rise to transient positive LD (Figure 4), but it would seem that many situations would also give rise to negative LD. Is there a symmetry to the model such that if you were to "integrate" across all possible starting conditions, the expectation would be zero?

In this example, the shared dependence of the additive selection coefficients upon S^x^ means that there is a tendency to produce positive (metapopulation) LD, both transiently and at equilibrium. In order for this model to produce negative LD, we would need there to be some (strong) factor generating negative LD to counteract this effect. One possibility would be negative epistasis, but this needs to be very strong, as it needs to overcome the positive covariance between additive selection coefficients induced by the shared dependence on S^x^.

In terms of the initial conditions, we are starting from a state in which there is no LD, and then changing the selective conditions faced by the metapopulation. This then leads to alleles segregating within the population in response to the new selective pressures, which in turn generates LD.

If we were to instead manipulate the initial conditions (i.e., set up negative LD or positive LD) without changing the selective pressures, this is a slightly different problem and is related to the Hill Robertson effect in which LD can interfere with directional selection. In this case, HR theory predicts negative LD would persist longer in the population than positive LD by weakening directional selection. We now discuss HR effects in more detail in the conclusion.

Or is there an asymmetry that makes transient LD more likely to be positive? I assume the latter if you add epistasis, but it's unclear with additive selection. This would help map intuition from your model to results such as Figure 3, in which it seems there's a skew towards positive LD. Is epistasis absolutely needed to explain this skew?

We detailed the emergence of LD at the metapopulation and at the population scale in the revised version. At the metapopulation scale, the positive LD is due to the heterogeneity in baseline growth rate among serotypes (our equation 7 shows how this leads to D_M_>0). At the scale of serotypes, we find that, indeed, epistasis is required to generate LD (see figure 5e and 5f).

2) One interesting implication of this work is what to expect when bacterial "populations" are either not defined at all or even incorrectly defined. Since Dtot is very positive under many scenarios (even with negative epistasis! Figure 4), you will observe an excess of positive LD between resistance elements (given these starting conditions in Figure 4, that is).

Yes. We like this comment regarding the fact that we often lack a proper description of the metapopulation structure of microbial communities. We now develop this idea further in the conclusions section. Thanks.

Reviewer 2:This is an interesting and important topic and the paper promised to be a new take on it, from the point of view of dynamics of linkage disequilibrium.My concerns are (1) clarity of the mathematics, notation and model development

We reworded several sections and changed the notation to improve the readability of our work. As pointed out above, we extended our discussion of the results in the examples section. We believe this analysis helps to show how our analytical work (equations 3 to 7) can help understand the complex dynamics.

and (2) with the claims that the model explains data, without a wide range of simulations or a direct comparison of model to data (for example see below re line 239 onwards).

We claim that our analysis can help to understand observed patterns. For instance, we provide a general understanding on the conditions promoting MDR. This analysis clarifies the link between heterogeneities among mechanisms of clearance and the evolution of drug resistance.

But we do not try to fit quantitatively observed patterns. This would be feasible in principle (parameter estimation) but, in the present work, our main objective was to present a qualitative understanding of the interplay between selection, migration and recombination.

The model in Equation (2) looks linear and the reader has to see figure 1 to note that there is actually a nonlinear term here (the susceptible * infectious term, and a product of two infectious terms) because the s_A_ etc depend on the prevalence.

We clarified that (line 66-67).

The model derivation is cumbersome and far from straightforward; I was unable to directly derive (3) from (2). Instead, for the first three terms of (3) I obtaineddpA/dt = (1-pA)(pA sA + pAB (sAB – sA – sB) ) + sB Dinstead of the expression in the paper which is(1-pA) (pA sA + sAB pAB) + sB DThis would seem to carry through to the other equations. However my derivation attempts were simply using sA, sAB, sB as given in the text, and putting in the rate of growth of the AB variant (r + sAB) as in (2) for example, without substituting in the expressions from Figure 1 for sA, sB, and sAB. This may work it out (?) However this left me a bit sceptical.

We have changed the notation to avoid this type of confusion. s_AB_ was meant to describe epistasis in fitness (not selection on AB genotypes; which is s_A_ + s_B_ + s_AB_). We now use the notation s_E_ for epistasis in fitness, and we are very explicit in the main text and in the appendix how we derive the different equations.

Bottom of page 4: if doubly-resistant infections are overrepresented in the population, D^X^ > 0. < -- middle page 5 "since D^X^ > 0" Which is it?

We have clarified the wording in this paragraph.

Page 10: "…, drug B resistance is often more likely to occur in an infection with a genetic background resistant to drug A." -- more likely than what?

We have rewritten this entire example and so this sentence no longer appears.

Line 239++ : This section makes the claim:"Thus transient dynamics coupled with epistasis can explain the significant within-serotype LD observed in Streptococcus pneumoniae".This is a big claim, based on a qualitative paragraph with no direct comparison to data, in which there is an underlying assumption of fixation of resistance to drug A in many serotypes. To my knowledge we often do not see fixation of resistance in Streptococcus pneumoniae or in the Maela data in particular, but rather long-term coexistence. I am not convinced that the dynamics of this model explain the significant LD in the data as claimed.

It was not our intent to claim that this simple example can fully explain the data. Our intention was to demonstrate that our dynamical equations help understand what we observe in the simulations. This is key because, at first sight, multilocus models may seem overwhelmingly complex. We show with this example that our analysis helps to understand both the short-term and long-term dynamics of drug resistance.

Note as well, that for the parameter values we have chosen (taken when possible from the Lehtinen et al., 2019 paper for Streptococcus Pneumonaie), it is straightforward to generate scenarios (as in Figure 5) in which there is a considerable amount of transient polymorphism with or without epistasis in figures 5a, b and c. The point here is that things which look like they are in equilibrium based upon surveillance data need not be.

Moreover, the existing explanation for the Maela data presented in Lehtinen et al. 2019 makes a very clear and testable prediction (as we discuss in the text): there should be no LD within-serotype. Yet this prediction is not met in the empirical data (see Figure 4). This would suggest a number of possibilities, including that either: (1) the populations are in equilibrium, but the coexistence mechanism proposed by Lehtinen et al., is not operating, or (2) the coexistence mechanism proposed by Lehtinen et al. is in effect, but the population is not in equilibrium.

In our paper we focus upon examining whether relaxing the conditions in (2), i.e., that the population is in equilibrium, could plausibly explain the patterns of LD in combination with the coexistence mechanism proposed by Lehtinen et al. We show that it could. We stress that this need not be the only explanation (e.g., condition (1) may be at play), and more work would be needed to definitively answer this question.

However, we recognize that the language we previously used in the manuscript may have been misleading about what we were and were not trying to do, and what our analysis does and does not say. As such, we have carefully gone through the manuscript to ensure that the language is not misleading about any conclusions of our analysis, and we have been more specific about what our objectives were for this example.

Equation (2) – there is a note above that rho_l_^X^ depends on infection densities; the nature of this dependence should be specified.

We have added a reference on line 52-53 indicating the equation in the Sup. Mat. which specifies the full dependencies.

Page 12: why, if there is no mechanism maintaining within-population diversity, does this necessarily mean D^X^ = 0 (line 282) ?

This sentence specifically refers to equilibrium in a deterministic model consisting of independent populations. If nothing maintains within-population diversity, then some combination of alleles will be fixed in population $x$, e.g., $f^x^_Ab_ = 1$ (using the notation where “f_ij_” refers to the frequency of genotype ij). Since $D^x^ = f_AB_^x^ f_ab_^x^ – f_Ab_^x^ f_aB_^x^$, when $f^x^_Ab_ = 1 $and so $f_AB_^x^ = f_ab_^x^ = f_aB_^x^ = 0$, then it follows that $D^x^ = 0$.

[Editors’ note: what follows is the authors’ response to the second round of review.]

Essential Revisions:While the reviewers of the previous version of your paper were not available to reconsider the revised manuscript, the reviewers of this version of the paper did take the previous comments and your responses into consideration when composing their own reviews.Collectively we see much potential in your manuscript, but we request the following points to be addressed at minimum (see below reviews for further details and more suggestions):1. Expanded explanation of the assumptions behind the LD framework. In particular:a) Is epistasis necessarily defined in terms of an additive expectation on growth rate? (If this is a standard result in population genetics, a reference to an accessible explanation would be enough).

Yes, for continuous time models epistasis is defined in terms of additive expectations on growth rate. This is because in continuous time models, fitness is per-capita growth rate, *r*. For discrete time models, epistasis is multiplicative (see e.g., Sup. Info. of Kouyos et al., 2009 Epidemics). We have added a paragraph stating this on lines 70-78.

But to see why it is a necessity, denote per-capita growth of genotype (strain) *jk* ∈ {*ab,Ab,aB,AB*} in population *x* as rjkx. Consider evolution restricted to locus *A* when allele *b* is fixed, and let pAx be the frequency of allele *A* in population *x*. Then the evolution of allele *A* is governed by

dpAxdt=(rAbx−rabx)pAx(1−pAx) Therefore allele *A* will increase in frequency whenever rAbx>rabx; thus sAx=rAbx−rabx is the selection coefficient for allele *A*. Now epistasis is defined as any interaction between loci affecting fitness. Since sAx and sBx are the additive contribution to baseline fitness, rabx , by strains carrying alleles *A* or *B*, the fitness of strain *AB* is the ‘baseline’ fitness, plus the change due to carrying allele *A* and *B* (sAx and sBx, respectively), plus any remaining fitness unaccounted for due to interactions between loci, sEx (epistasis): rABx=rabx+sAx=sBx+sEx Rearranging gives sEx=rABx−rabx−sAx−sBx=rABx+rabx−rAbx−raBx Thus epistasis in fitness is defined in terms of the per-capita growth rates.

There are many different ways we can model the costs of carrying resistance. All are a priori valid but what we want to stress here is the consequences of the way we model these costs on epistasis and consequently on LD.

For a generic epidemiological model, let *β^x^S^x^* and *µ^x^
*be the per-capita transmission and mortality rate in population *x*, where *S^x^
*is the available density of susceptibles. Let cβkxSx and cμkx be the decrease in transmission and increase in clearance due to carriage of the resistance allele *k*. Finally, let sEx denote any epistatic interactions.

Then the per-capita growth of strain *jk* is

rjkx=(βx−1A(j)cβAx−1B(k)cβBx)Sx−1a(j)τBx−1b(k)τBx−μx−1A(j)cμAx−1B(k)cμBx+1A(j)1B(k)sEx, where 1𝓁(𝓏) is 1 if 1=𝓏 and zero otherwise. Therefore skx=τkx−cβkxSx−cμkx,k∈{A,B} For example, in Lehtinen et al., they de_ne_ per-capita growth of strain *k* in population

*i*, resistant to the set of antibiotics *R_k_* and sensitive to the set *S_k_* as

rkx=βx∏jϵRkcβjSx−μx∏jϵRkcμj−∑jϵSkτjx We have slightly modified their notation here, but the gist is the same; we have also ignored the possibility of combination treatment (but see Sup. Mat. 4.1 for an example in which it is included). Therefore in the case of two antibiotics, *A* and *B*,

we have cβkx=(1−cβk)Sx and cμkx=(cμk−1) and so

skx=τkx−(1−cβk)βxSx−(cμk−1)μx,k∈{A,B} and sEx=(1−cβA)(1−cβB)βxSx−(cμA−1)(cμB−1)μx Thus there exists epistasis in fitness whenever there is a cost of resistance. More specifically, transmission costs will produce positive epistasis, while duration of carriage costs will produce negative epistasis. We have added further clarification on lines 716-738.

b) The s coefficients are dependent on variable densities – does this matter for the partitioning and interpretation of equations 3 and 4?

The partitioning and interpretation of equations 3 and 4 are unaffected by the *s* coefficients depending upon variable densities; the *s* coefficients have to be partitioned as they are from consideration of the single locus dynamics (see equation (1)); and this then forces epistasis to be what it is.

Yet, evolutionary dynamics can be more complicated in evolutionary epidemiology than in classic population genetic models as epidemiological feedbacks can change the sign and magnitude of the selective coefficients and epistasis. Although this creates richer dynamics, the fundamental population genetic principles still apply, and provide a useful interpretative framework.

2. If the authors disagree with Reviewer 1's questions about the interpretation that variation in susceptible density explains the effect in example 1 and in Lehtinen et al. 2019, more explanation is needed for the following points:a) Why variation in clearance rate affects selection for resistance when there is a multiplicative cost on clearance rate, but no cost on transmission rate (for the re-interpretation of Lehtinen specifically)?

Consider the evolution of resistance to drug *A* in a metapopulation in which allele *b* is fixed. From (1), resistance is selected for in population *x* whenever the selection coefficient is positive, sAx>0. So resistance is selected for whenever the benefits (τAx) exceed the costs (cβAxSx+cμAx). This is true when the system has reached an endemic equilibrium but also during transient dynamics.

Now how is variation between the populations generated? This could occur a number of ways; but in all cases, the key is that the selection coefficient, sAx, differs between populations. For example, variation in treatment rate or in the parameters controlling the costs are sufficient. However, in keeping with Lehtinen et al., 2017, 2019, we assume that τAx=τA and that the cost parameters are population independent.

Then there are two ways in which variation can be produced, depending upon if the cost parameters are additive or are multiplicative (i.e., do they interact or not with the epidemiological parameters?)

i) Additive costs (cβAx=cβA,cμAx=cμA) : here costs are ‘independent’ of the population, except insofar as the transmission costs also depend upon density of susceptibles.

From sAx, variation in resistance (due to selection) across populations is only possible if both *c_βA_ >* 0 and the susceptibles, *S^x^*, show variation.

This was the example we focused upon in the manuscript; the prediction here is that any factor negatively correlated with *S^x^
*will lower the costs of resistance and so select for resistance. For example, populations with a longer duration of carriage (smaller *µ^x^
*or longer duration of carriage), or higher transmissibility (larger *β^x^*), will more substantially deplete available susceptibles, *S^x^*, lowering the costs of resistance, cβASx+cμA, and so select for resistance.

ii) Multiplicative costs: here the cost parameters multiplicatively interact with the epidemiological parameters. This was true in the model of Lehtinen et al., 2017, 2019 (detailed above); for this model, by assumption (due to the multiplicative interaction), populations with a longer duration of carriage (smaller *µ^x^*) pay disproportionately lower costs to acquire resistance than populations with shorter duration of carriage (larger *µ^x^*), and so are more likely to become associated with resistance.

Another way of looking at this is if we suppose the populations correspond to serotype, then by specifying an interaction between cμA and *µ^x^*, we are creating epistasis between population (serotype) *x* and resistance. Epistasis produces LD (in this case between serotype and resistance).

Therefore there are two distinct mechanisms generating variation: through the density of susceptibles, or by assuming multiplicative interactions between the cost parameters and the epidemiological parameters. We focused upon additive costs, because we wanted to emphasize the correlation between selection coefficients due to susceptible densities, but this does not mean that multiplicative costs are not valid.

For example, in the case of *S. pneumoniae*, there may be a physiological reason why the mechanism of resistance interacts with capsule, such that certain capsule types reduce the efficacy of antibiotic resistance over others, or make resistance more costly. Since different capsule types are associated with different durations of carriage, it is possible that such an interaction might mean that capsules with shorter duration of carriage amplify the costs of resistance.

In the Sup. Mat. on lines 716-738 we have clarified this issue; we have also modified the language in the main text (lines 247-250, lines 290-292) when we are discussing the Lehtinen et al., example to make clear that we are primarily focused upon correlations in variation mediated through susceptibles.

b) What is happening in example 2, where the effect is currently explained in terms of serotype-specific susceptible density, but the model does not include serotype-specific susceptibles?

The pool of available susceptibles depends upon serotype due to the stabilizing function intended to mimic adaptive immunity (this is the function from Lehtinen et al., 2017). So the *ν*(*I,x*) generates the fraction of susceptibles that are available to serotype *x*, i.e., *S^x^
*= *ν*(*I,x*)*S*. Thus there is variation in available susceptibles between serotypes and so there is serotype-specific susceptible density.

It would be possible to mechanistically model each of the populations of susceptibles, however, this would be computationally difficult and not add anything, as the stabilizing function ensures that there is: (1) the potential for serotype-specific differences in available susceptibles and (2) the variation in available susceptibles is linked to attributes of the serotypes (specifically, fitness).

In the Sup. Mat. on lines 756-763 we have added text which addresses this issue.

3. For examples 2 and 3, please either explain why additive transmission costs are reasonable (given the concerns highlighted below) or change the modeling of cost.

‘Additive’ transmission costs are always reasonable, given suitable constraints (e.g., ensuring transmission cannot be negative, as in the example provided by the reviewer); but the requirement for suitable constraints is also true for ‘multiplicative’ costs (e.g., the constraint 0 ≤ *c_β_
*≤ 1). In the analysis we conducted in examples 2 and 3, appropriate constraints were attached. The key distinction is the costs have different evolutionary implications (as reviewer 1 correctly pointed out).

More generally, we have reframed the relevant sections so it does not seem as if we are advocating for either additive or multiplicative costs; our intent was to point out that they have different evolutionary interpretations. For example, one consequence of multiplicative costs is that they tend to produce epistasis.

We also have amended the text to stress that the inclusion of epistasis in a model is not ‘problematic’ (as our wording incorrectly suggested in the Sup. Mat.); in fact, in many circumstances it is necessary. Our point was that awareness of the presence/absence of epistasis can guide the interpretation.

[Editors’ note: further revisions were suggested prior to acceptance, as described below.]

The manuscript has been improved but there are some remaining issues that need to be addressed as outlined below in edited comments from Reviewer Sonja Lehtinen, as outlined in points 2-5 below. In addition, I have reviewed the discussion with the eLife editorial office about the article structure and have considered this question myself, ultimately coming to point 1.1. Please do create a methods section for this paper. There is no need to remove anything from your current main text. Rather, in the new methods section please first overview the methods even if they are already described in more detail elsewhere in the paper (some repetition is ok, and I agree with the level of detail you have provided in the Results section, given the type of paper) in one or more subheadings, and then import all of the sub-headings and the full content of the appendix. Since we don't have page limitations and since this is important to the paper, I think it should be included in the methods section of the main text of the paper, rather than as an appendix.

We have now created a methods section that is the appendix. We have added a paragraph on lines 589600 providing an overview of the contents of the methods. In the main text, we now have changed all the references to refer to the relevant sections in the methods.

2. The point the authors are making about density of susceptibles and that this point is not limited to additive costs is understood. However, the section on equilibrium patterns of MDR still should be updated because the difference between additive and multiplicative costs is not explicitly flagged in the main text. (I think the discussion in the Sup Mat is very helpful, but most readers will not read the Supplement). Specifically, when considering equilibrium patterns of MDR, the paragraph starting line 283 is only true for additive costs. Without reading the supplement, this paragraph comes across as a general result. If costs are multiplicative, variation in duration of carriage is both necessary (i.e. variation in transmission rate does not produce LD at equilibrium) and sufficient (explicit transmission costs are not needed). Thus, overall, it is true to say that variation in duration of carriage is not necessary, but not that it is not sufficient. It would be helpful to explicitly make this difference clear somewhere in this section. Please also rephrase the sentence lines 270-273 to avoid implying that the costs in Lehtinen et al., are additive.

We have made a number of changes to address these points. First, we have added a box to the main text that discusses the costs of resistance (specifically, additive vs multiplicative) and their consequences for epistasis on pg. 5.

Next, we have made a number of changes to the wording to clarify the points the reviewer is concerned about. Specifically:

– On line 251, we identify that we are using additive costs

– On line 252 we state that this differs from Lehtinen et al., 2019 who use multiplicative costs

– On line 276-277 we identify again that our result here differs from Lehtinen et al., 2019 based upon the choice of costs

– On line 287 we state that our results are for additive costs

– In the caption for figure 3 we identify that we are dealing with additive costs (and no epistasis)

3. The main text would benefit from a discussion of additive costs. Specifically:a. The authors make a good point that cost parameters are subject to constraints however they are modelled. The reviewer's point was that these constraints are qualitatively different for additive transmission costs: reasonable additive costs for single resistance can lead to negative transmission rate for dual resistance (as in the example in my review) in the absence of cost epistasis. This suggests that for cost parameters where this is the case, assuming no epistasis cannot be appropriate. Although the specific parameter values the authors use don't give rise to this problem, the authors also present more general results for a model with additive costs and no epistasis, so it would be helpful to highlight this constraint somewhere.

In box 1 we have highlighted the constraint, it is that $0 \le \β_ij_^x^ \le \β_ab_^x^$. This constraint applies to any form of cost.

In the case of additive costs, there are a variety of ways this constraint could be implemented. We discuss this on lines 762-772 and provide two examples – one of which can lead to epistasis.

b. The authors' careful consideration of how the specification of costs affects interpretation is indeed very useful and interesting. In addition to this conceptual point, the authors say that they are aiming to highlight mechanisms – e.g. variation in transmission rate – that could plausibly give rise to patterns of MDR. In the case of equilibrium patterns, variation in transmission rate only gives rise to LD when transmission costs are additive. The plausibility of variation in transmission rate as an explanation for equilibrium patterns of MDR therefore depends on the plausibility of additive transmission costs. Therefore, if the authors want to suggest that variation in transmission rate is a plausible explanation for patterns of MDR, it is necessary to include a discussion of whether/how additive transmission costs might arise.

The magnitude of the costs and epistasis will be dictated by the biology of the system. For example, when discussing populations separated based upon capsular serotype, the structure of the capsule has a role in determining transmission/duration of carriage. If the resistance mechanism does not interact with these differences in capsule structure, then carriage of the resistance allele will come with additive costs. However, it is also possible that the differences between capsule serotype may interact with the resistance mechanism. For example, more transmissible capsular serotypes may pay lower or higher costs of resistance. If more transmissible serotypes pay higher costs, this is multiplicative costs (or at least how we have been using it); if more transmissible capsular serotypes pay lower costs, this would require a slightly different form of the costs – but this would also yield epistasis. Similar logic applies to duration of carriage. We have added a section in the Methods (lines 754-761) to address this.

More generally, without specific evidence of an interaction between differences between capsular serotypes and the resistance mechanism, then a priori it is reasonable to assume there is no interaction (additive costs).

4. Add "per carriage episode" to the summary of the biological interpretation of the duration of carriage effect in Lehtinen et al., 2017/2019 (lines 244-245), to make it clear that they were not suggesting that (host) populations with longer durations of pathogen carriage have greater antibiotic exposure. (This point was missed in the original review).

Whoops, yes we have added that. Thanks for pointing it out.

5. One final observation, which is not an essential revision and the authors should only address it if they think it will improve the paper. Points 2 and 3 are both related to the result that a model with a multiplicative transmission cost predicts, at equilibrium, LD between duration of carriage and resistance, but not transmission rate and resistance. Would the authors explain this in terms of variation in the density of susceptibles giving rise to the LD with duration of carriage, but this effect being offset by the epistatic interaction between transmission rate and cost of resistance leading to no LD between transmission rate and resistance? This might be interesting to explicitly discuss in the manuscript.

If we understand this scenario correctly, the reviewer is suggesting a model in which the selection coefficients are of the form :

$s_d_^x^ = -c_{\β_d}^x^\β^x^S^x^ + \tau_d_^x^$ and there is variation in duration of carriage (but no costs of resistance to duration of carriage).

In this model, there is no way to isolate the susceptible density so that it is the only thing in the selection coefficient varying across populations without also setting $\β^x^ = \β$. This is because the parameter controlling the `costs’ in this case is not just $c_{\β_d}^x^$, but is actually the composite parameter $c_{\β_d}^x^\β^x^$. This is related to our point about the interpretation of multiplicative costs: they assume a dependence upon other life-history characteristics.

At equilibrium, as the reviewer pointed out, there is cancellation between the $\β^x^$ term in the costs and the $\β^x^$ term in the denominator of the susceptible density. But this is really just a product of the mathematical assumptions; and it would be incorrect to refer to c_{\β_d} as the `costs’ because the cost `parameter’ actually depends upon $\β^x^$.

Put another way, in the most general setting we could have simply stated we have transmission rates \β_ij_^x^ without specifying any relationship whatsoever between `costs’ and ‘transmission’; in this case it is easy to see that only if we make specific mathematical assumptions will cancellation occur.

For example, if we were interested in a scenario in which populations with higher transmission rates pay disproportionately higher costs of resistance (which is the same biologic assumption as multiplicative costs), we would be equally justified in choosing the function

\β_{Ab}^x^ = \β^x^/(1 + c_{\β_A} \β^x^) (*)

as we would be in choosing multiplicative costs

\β_{Ab}^x^ = (1 – c_{\β_A})\β^x^ (#)

But using the function (*), at equilibrium \β^x^ won’t disappear from the invasion condition as it will with the function (#).

We have now highlighted in Box 1 that costs need not necessarily be ‘additive’ or ‘multiplicative’; they could just as easily take a more complex form as in our example (*) above. We simply want to emphasize that the choice of costs can produce epistasis, which is an important consideration for modelling multilocus dynamics.